# 1-BIT LAMB: COMMUNICATION EFFICIENT LARGE-SCALE LARGE-BATCH TRAINING WITH LAMB'S CONVERGENCE SPEED

## ABSTRACT

To train large models (like BERT and GPT-3) on hundreds of GPUs, communication has become a major bottleneck, especially on commodity systems with limited-bandwidth TCP network. On one side large batch-size optimization such as LAMB algorithm was proposed to reduce the frequency of communication. On the other side, communication compression algorithms such as 1-bit Adam help to reduce the volume of each communication. However, we find that simply using one of the techniques is not sufficient to solve the communication challenge, especially under low network bandwidth. Motivated by this we aim to combine the power of large-batch optimization and communication compression, but we find that existing compression strategies cannot be directly applied to LAMB due to its unique adaptive layerwise learning rates. To this end, we design a new communication-efficient algorithm, 1-bit LAMB, which introduces a novel way to support adaptive layerwise learning rates under compression. In addition, we introduce a new system implementation for compressed communication using the NCCL backend of PyTorch distributed, which improves both usability and performance. For BERT-Large pre-training task with batch sizes from 8K to 64K, our evaluations on up to 256 GPUs demonstrate that 1-bit LAMB with NCCL-based backend is able to achieve up to 4.6x communication volume reduction, up to 2.8x end-to-end time-wise speedup, and the same sample-wise convergence speed (and same fine-tuning task accuracy) compared to uncompressed LAMB.

## 1 INTRODUCTION

Training large-scale deep learning models in a distributed fashion is computation-heavy and expensive (Brown et al., 2020). In addition to computation, communication overhead becomes a serious system challenge for such large models. A recent study of BERT pre-training with Adam demonstrates that the allreduce communication can take up to 94% and 75% of total training time per step on clusters with Ethernet and InfiniBand inter-node connections, respectively (Tang et al., 2021).

To achieve communication efficient distributed training, there are two promising directions: large batch optimization and communication compression. **LAMB** optimizer, which can be viewed as Adam with adaptive layerwise learning rates, is an example of large batch optimization (You et al., 2020). LAMB can scale the batch size of BERT pre-training to 64K without losing accuracy, thereby greatly reducing the total training time as larger batch sizes leads to less frequent communication. On the other hand, recent works on communication compression such as **1-bit Adam** demonstrate that it is possible to combine 1-bit compression with Adam's convergence speed, thereby reduce BERT pre-training communication volume by 5x (Tang et al., 2021).

Both LAMB and 1-bit Adam demonstrate great benefit for distributed training. Unfortunately, our studies show that simply using one of them is not sufficient to fully address the communication issue, especially under limited network bandwidth and large number of GPUs/machines (Section 3). We find that communication is still a non-trivial overhead when running large-scale distributed training with LAMB, even with the larger batch sizes. Previous study shows that Adam provides slower convergence speed compared to LAMB at batch sizes 16K or larger for BERT pre-training (You et al., 2020). Using the same methodology, our BERT experiments show that 1-bit Adam, similar

to Adam, also has slower convergence speed compared to LAMB at batch size 16K. Even with the communication compression, this batch size limitation would hurt the communication efficiency when the number of GPUs/machines is large.

LAMB and 1-bit Adam are two unique optimizers. However, the techniques behind them are complementary: large batch optimization reduces the frequency of communication, and compression reduces the volume of communication. Motivated by this we aim to combine LAMB's large batch optimization algorithm with compression strategies behind 1-bit Adam. However, we find that they are not directly compatible due to LAMB's unique layerwise learning rate update strategy, which requires information that are missing when communication and optimizer states are compressed (Section 3).

The studies and challenges above motivate us to design a new algorithm called **1-bit LAMB** (Section 4). Learning from the insights behind 1-bit Adam, 1-bit LAMB is a 2-stage algorithm which uses LAMB (warmup stage) to "pre-condition" a communication compressed momentum SGD algoirthm (compression stage). At compression stage where original LAMB algorithm cannot be used to update the layerwise learning rates, 1-bit LAMB employs a novel way to adaptively scale layerwise learning rates based on information from both warmup and compression stages. As a result, 1-bit LAMB is able to achieve large batch optimization (LAMB)'s convergence speed under compressed communication, which is impossible using existing approaches.

In addition to the 1-bit LAMB algorithm, we propose a new NCCL-based compressed communication backend which provides better usability and performance than previous work (Section 5). This backend can be applied to 1-bit LAMB, 1-bit Adam, and other communication compression algorithms. We evaluate 1-bit LAMB using BERT pre-training and GLUE/SQuAD fine-tuning tasks (Section 6). Results show that under different batch sizes from 8K to 64K and with up to 256 GPUs, 1-bit LAMB with NCCL-based backend is able to achieve up to 4.6x communication volume reduction and up to 2.8x end-to-end time-wise speedup for BERT pre-training compared to uncompressed LAMB, together with the same sample-wise convergence speed and same GLUE/SQuAD fine-tuning task accuracy. The 1-bit LAMB optimizer as well as the NCCL-based communication backend has been open sourced in a deep learning optimization library (name hidden to maintain anonymity).

## 2  RELATED WORK AND BACKGROUND

To achieve communication efficient distributed training, techniques include decentralization (Lian et al., 2017; Koloskova* et al., 2020; Li et al., 2018), asynchronous communication (Zheng et al., 2016; Chaturapruek et al., 2015), and gradient compression/quantization which we focus on in this paper. Before communication, we could compress the original gradient $g$ into $\mathcal{C}_\omega[g]$, where $\mathcal{C}_\omega[\cdot]$ is the compress operator[1]. As a result the communication volume could be greatly reduced. Compression can be achieved by quantization, sparsification, sketching, etc. (Ye & Abbe, 2018; Alistarh et al., 2017; Agarwal et al., 2018; Yu et al., 2019; Spring et al., 2019; Ivkin et al., 2019; Shi et al., 2021). Several works focus on unbiased compression methods (original and compressed tensors have the same expectation), such as centralized compressed parallel SGD (Alistarh et al., 2017) and many others (Wangni et al., 2018; Shen et al., 2018; Zhang et al., 2017; Wen et al., 2017; Jiang & Agrawal, 2018). On the other hand, recent works about biased compression methods demonstrate better compression rate and the same convergence rate by using an error compensation technique (Seide et al., 2014; Bernstein et al., 2019; Stich et al., 2018; Zheng et al., 2019; Phuong & Phong, 2020; Yu et al., 2019; Shi et al., 2019; Ivkin et al., 2019; Sun et al., 2019; Basu et al., 2019; Vogels et al., 2019; Tang et al., 2021).

The error-compensated compression is proposed in the 1-bit SGD work (Seide et al., 2014): instead of compressing the gradient at each iteration directly, they compress the sum of the gradient and the last step's compression error. By using error compensation the training can achieve promising convergence speed even with 1-bit compression (representing the gradient by $\pm 1$ signs and a scale). Recent works provide theoretical guarantee of this method (Bernstein et al., 2019), and also demonstrate that it admits the same asymptotic convergence rate as the uncompressed one (Stich et al., 2018). In addition, error compensation method enables almost any compression methods (Stich et al., 2018), either biased or unbiased, to converge as fast as the uncompressed case.

---

[1] $\mathcal{C}_\omega[\cdot]$ could also include randomness.

**Adam** (Kingma & Ba, 2015) can be viewed as SGD with momentum and adaptive learning rate scaling on each coordinate of the gradient. It has demonstrated promising convergence speed and hyperparameter robustness on many deep learning tasks. Recently, Tang et al. (2021) proposed **1-bit Adam** which combines the efficiency of error-compensated 1-bit compression with Adam's convergence speed. They show that error-compensated compression does not work for Adam directly, because Adam is non-linearly dependent on the gradient (the variance term). On the other hand, they find that Adam's variance becomes stable at an early stage of training. To this end, they design a new 2-stage algorithm, 1-bit Adam: At warmup stage, vanilla Adam is used. At compression stage, they stop updating the variance and use it as a fixed precondition, and communicate based on the momentum applied with error-compensated 1-bit compression. Their experiments on up to 256 GPUs show that 1-bit Adam achieve the same convergence behaviour and final accuracy as Adam, together with up to 5x less communication volume and 3.3x faster end-to-end throughput.

To further improve training efficiency at large scale, being able to support large minibatches while keeping the convergence speed is a critical factor. Recently You et al. (2020) find that it is difficult to keep Adam's convergence speed at batch sizes 16K or larger for BERT pre-training. To this end they proposed **LAMB** which can be viewed as Adam with adaptive layerwise learning rates. By using LAMB, they are able to scale the batch size of BERT pre-training to $64K$ without losing accuracy, thereby, reducing the BERT training time from 3 days to around 76 minutes. The major idea of LAMB is that it utilizes a layerwise scaling coefficient to regulate the update of each layer, and the updating rule can be summarized as[2]:

$$
\begin{aligned}
\boldsymbol{m}_t^{(l)} =& \beta_1 \boldsymbol{m}_{t-1}^{(l)} + (1-\beta_1)\boldsymbol{g}_t^{(l)}, \boldsymbol{v}_t^{(l)} = \beta_2 \boldsymbol{v}_{t-1}^{(l)} + (1-\beta_2)(\boldsymbol{g}_t^{(l)})^2 \\
\boldsymbol{u}_t^{(l)} =& \frac{\boldsymbol{m}_t^{(l)}}{\sqrt{\boldsymbol{v}_t^{(l)}} + \eta}, c_t^{(l)} = \text{clip}\left(\frac{\|\boldsymbol{x}_{t-1}^{(l)}\|}{\|\boldsymbol{u}_t^{(l)}\|}, c_{min}, c_{max}\right) \\
\boldsymbol{x}_t^{(l)} =& \boldsymbol{x}_{t-1}^{(l)} - \gamma c_t^{(l)} \boldsymbol{u}_t^{(l)}.
\end{aligned}
\tag{1}
$$

Here $\boldsymbol{g}_t^{(l)} = \nabla F(\boldsymbol{x}_t; \boldsymbol{\xi}_t)$, $\boldsymbol{m}_t^{(l)}, \boldsymbol{v}_t^{(l)}, \boldsymbol{x}_t^{(l)}$ denote the stochastic gradient, momentum, second moment (i.e., the variance), and the model parameters at the model's $l$-th layer at step $t$; $\beta_1$ and $\beta_2$ are the decaying factor; $\gamma$ is the learning rate; $\eta$ is an additive constant to avoid division by 0; $\text{clip}(x, a, b) := \min\{\max\{x, a\}, b\}$ is the clipping operation[3]; $c_t^{(l)}$ is a layer-wise scaling factor that regulates the update of $\boldsymbol{x}_t^{(l)}$ into certain range. One thing to note is that within each layer, each tensor (e.g., weight and bias) will have its own scaling coefficient $c_t^{(l)}$. The underlying intuition of LAMB's scaling coefficient is that when the update is relatively large compared to the parameter, we should apply a lower learning rate to that layer (and vice versa).

## 3 MOTIVATION AND INSIGHTS

### 3.1 1-BIT ADAM IS NOT SUFFICIENT FOR LARGE-BATCH DISTRIBUTED TRAINING

1-bit Adam demonstrates the same convergence speed as Adam for BERT pre-training task with batch size 4K (Tang et al., 2021). On the other hand, the LAMB work shows that it is difficult to keep Adam's convergence speed at batch sizes 16K or larger for BERT pre-training (You et al., 2020). To find out whether 1-bit Adam is sufficient for large-batch distributed training, we perform a similar experiment using BERT pre-training task at batch size 16K. Using You et al. (2020)'s training parameters and tuning procedure (details in Appendix A.1) for LAMB and Adam, we perform BERT pre-training with LAMB and 1-bit Adam, respectively. Then we use the two pre-trained BERT model to perform SQuAD 1.1 fine-tuning (details in Section 6). Results in Table 1 show that similar to Adam, 1-bit Adam has slower convergence speed compared to LAMB at larger batch size.

---

[2]Here $(\boldsymbol{x})^2$, $\sqrt{\boldsymbol{x}}$ and $\frac{x}{y}$ all denote element-wise operations. For simplicity weight decay is omitted.

[3]In the LAMB paper the clip function is only applied to $\|\boldsymbol{x}_{t-1}^{(l)}\|$ without mentioning the exact clipping function configurations, and our experiments show that $\|\boldsymbol{x}_{t-1}^{(l)}\|$ varies a lot among different layers. Thus we apply the clipping function to the whole ratio, which is more stable among different layers. With this clipping function we are able to achieve similar SQuAD accuracy compared to the original LAMB.

Table 1: BERT-Large pre-training (batch size 16K) final validation loss, and SQuAD average/max dev set F1 scores over 32 runs using the pre-trained BERT models. The first two columns are from the original LAMB work. The last two columns are our experiments using the same training parameters.

| BERT pre-training optimizer | LAMB (You et al., 2020) | Adam (You et al., 2020) | LAMB | 1-bit Adam |
|---|---|---|---|---|
| BERT validation loss | — | — | 1.362 | 1.504 |
| SQuAD Avg. F1 | — | — | 90.716 | 89.225 |
| SQuAD Max F1 | 91.345 | 88.540 | 91.119 | 89.494 |

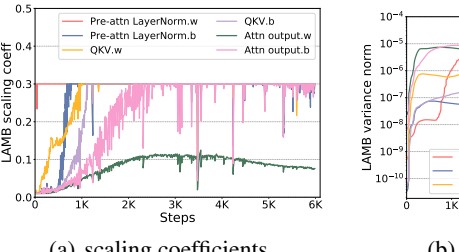 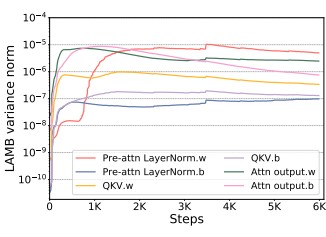

(a) scaling coefficients        (b) variance norms

Figure 1: LAMB's scaling coefficients ($c_t^{(l)}$ in (1)) and variance norms for different layers in the first BertLayer during BERT-Large pre-training seqlen 128 (5993 steps in total). We set the lower/upper bound of the scaling coefficient ($c_{min}$ and $c_{max}$ in (1)) at 0.01 and 0.3.

## 3.2 COMMUNICATION IS STILL AN OVERHEAD FOR LARGE-BATCH DISTRIBUTED TRAINING

Tang et al. (2021) demonstrate that when pre-training BERT with Adam at batch sizes 64 to 4K, the communication may take up to 94% and 75% of total training time per step on clusters with Ethernet and InfiniBand networks. To investigate whether the communication overhead still exists, we conduct a similar profiling experiments but with LAMB optimizer and batch sizes 8K to 64K. We evaluate two clusters: one with 4 NVIDIA V100 GPUs per node and 40 Gigabit Ethernet inter-node network (4.1 Gbps effective bandwidth); the other one with 8 V100 GPUs per node and 100 Gigabit InfiniBand EDR inter-node network (close to theoretical peak effective bandwidth). Results show that even with larger batch sizes the communication still contributes up to 91% and 52% of the training time on two clusters (details in Appendix A.2), indicating the opportunities to improve large-batch distributed training efficiency by communication compression.

## 3.3 INVESTIGATING BERT PRE-TRAINING WITH BASELINE LAMB

Tang et al. (2021) find that Adam's variance term becomes stable at an early stage of training (after around 15% of total training for BERT), which is why 1-bit Adam can "freeze" the variance after the warmup stage and use it as a fixed precondition during compression. LAMB also has the variance term as a non-linearly gradient dependency, and LAMB's scaling coefficient ($c_t^{(l)}$ in (1)) depends on the variance. Thus we investigate how LAMB's scaling coefficient and variance change during training using BERT pre-training task.

Figure 1 presents LAMB's scaling coefficients and variance norms for different layers in the first BertLayer (other layers in the model have similar patterns, details in Appendix A.3). Results demonstrate that the scaling coefficients keep increasing until reaching plateaus, because the update tends to become smaller compared to the parameter during training. In addition, many scaling coefficients become stable at an early stage. Results also demonstrate that LAMB provides adaptive learning rate in two folds: 1) different layers may reach scaling coefficient plateaus at different time; 2) different layers may reach different scaling coefficient plateaus. We believe that this is one of the reasons why LAMB can provide better convergence (or reach the same convergence with less hyperparameter tuning) at larger batch sizes compared with Adam. On the other hand, LAMB's varaince terms are less stable compared with Adam: many layers have their variance norms constantly decreasing during the whole training, up to two orders of magnitude difference. As we see in next section this makes 1-bit Adam's strategy affect convergence when directly applied to LAMB.

Table 2: BERT-Large pre-training (batch size 64K/32K at seqlen 128/512) final validation loss, and SQuAD average/max dev set F1 scores over 32 runs using the pre-trained models. "LAMB + basic 1-bit" is the experimental algorithm described in Section 3.4. " 1-bit LAMB" is the proposed work described in Section 4.

| BERT pre-training optimizer | LAMB (You et al., 2020) | LAMB (ours) | LAMB + basic 1-bit | 1-bit LAMB |
| --- | --- | --- | --- | --- |
| BERT validation loss | – | 1.451 | 1.494 | **1.443** |
| SQuAD Avg. F1 | – | 90.265 | 90.069 | **90.524** |
| SQuAD Max F1 | 90.584 | 90.555 | 90.409 | **90.788** |

### 3.4 EXISTING COMPRESSION STRATEGY AFFECTS LAMB'S CONVERGENCE

To combine 1-bit compression and large-batch training's communication efficiency, first we attempt to directly apply 1-bit Adam's strategy to LAMB. We design and evaluate an experimental two-stage algorithm "LAMB + basic 1-bit": At the warmup stage we use vanilla LAMB. At the compression stage we stop updating the variance and LAMB's scaling coefficient and use them as precondition, and communicate based on 1-bit compressed momentum (details about this algorithm design in Appendix A.4). For simplicity we only apply this experimental algorithm to BERT pre-training seqlen 128 phase, and still only use vanilla LAMB in seqlen 512 phase.

Table 2 presents the BERT pre-training final validation loss when using vanilla LAMB, the experimental algorithm described in this section, and the proposed 1-bit LAMB algorithm (Section 4). We also use the pre-trained models to fine-tune SQuAD 1.1 and present the F1 scores. Results show that simply freezing both variance and LAMB's scaling coefficient would affect the convergence speed and lead to lower SQuAD scores (and lower GLUE scores in Section 6). This is because LAMB's variance term is less stable as demonstrated in our study. On the other hand, the proposed 1-bit LAMB is able to provide the same convergence speed as vanilla LAMB, and next we will describe its algorithm design.

## 4 1-BIT LAMB ALGORITHM

The proposed 1-bit LAMB optimizer introduces a novel way to update the adaptive layerwise learning rate during the compression stage. There are two major differences between 1-bit LAMB and the original LAMB: 1) During compression stage, 1-bit LAMB updates the layerwise learning rate based on a novel "reconstructed gradient" based on the compressed momentum. This makes 1-bit LAMB compatible with error compensation and be able to keep track of the training dynamic under compression. 2) 1-bit LAMB also introduces extra stabilized *soft* thresholds when updating layerwise learning rate at compression stage, which makes training more stable under compression.

**Problem setting** In this paper, we focus on the following optimization task:

$$\min_{\boldsymbol{x} \in \mathcal{R}^d} \quad f(\boldsymbol{x}) = \frac{1}{n} \sum_{i=1}^{n} \underbrace{\mathbb{E}_{\boldsymbol{\xi}^{(i)} \sim \mathcal{D}_i} F(\boldsymbol{x}; \boldsymbol{\xi}^{(i)})}_{:=f_i(\boldsymbol{x})}, \qquad (2)$$

where $d$ is the dimension of the input model $\boldsymbol{x}$, $\mathcal{D}_i$ is the data distribution of individual data sample $\boldsymbol{\xi}^{(i)}$ on the $i$-th worker, $F(\boldsymbol{x}; \boldsymbol{\xi})$ is the loss function.

**Notations and definitions** Throughout this paper, we use the following notations:

- $\nabla f(\cdot)$ denotes the gradient of a function $f$.
- $f_i(\boldsymbol{x}) := \mathbb{E}_{\boldsymbol{\xi} \sim \mathcal{D}_i} F(\boldsymbol{x}; \boldsymbol{\xi})$.
- $\| \cdot \|_p$ denotes the $l_p$-norm for vectors and matrices. Notice $p = \infty$ means the infinity norm.
- $\boldsymbol{C}_\omega(\cdot)$ denotes the randomized compressing operator, where $\omega$ denotes the random variable. One example is the randomized quantization operator, for example, $\boldsymbol{C}_\omega(0.7) = 1$ with probability $0.7$ and $\boldsymbol{C}_\omega(0.7) = 0$ with probability $0.3$.
- $\sqrt{\cdot}$ denotes the square root of the argument. In this paper if the argument is a vector, then it returns a vector taking the element-wise square root.
- $(\boldsymbol{x})^2$ denotes the element-wise square operation if $\boldsymbol{x}$ is a vector.

- $\frac{a}{b}$ or $a/b$ denotes the element-wise division operation if both $a$ and $b$ are vectors and their dimension matches.

---

**Algorithm 1** 1-bit LAMB

---

1: **Initialize**: $x_0^{(l)}$, $m_0^{(l)} = 0$, $v_0^{(l)} = 0$, $c_{avg}^{(l)} = 0$ for each layer. Learning rate $\gamma$, initial error $\delta = 0$, number of total iterations $T$, warm-up steps $T_w$, three decaying factor $\beta_1$, $\beta_2$, $\beta_3$ for LAMB's momentum, variance, and scaling coefficient. $r^{(l)} = 1$, $r_{min}$, $r_{max}$, $r_{threshold}$ for 1-bit LAMB. We use 1-bit compression as the compressing operator, which means $C_\omega[x] = \frac{\|x\|}{\|\text{Sign}(x)\|}\text{Sign}(x)$ where $\text{Sign}(x)$ is the tensor that only keeps the sign of each element in the original tensor $x$.

2: Running the original LAMB in (1) for $T_w$ steps, and at each step $c_{avg}^{(l)} = \beta_3 c_{avg}^{(l)} + (1 - \beta_3)c_t^{(l)}$.

3: At the end of step $T_w$, for each layer store the variance term (defined as $v_t^{(l)}$ in (1)) $v_{T_w}^{(l)}$ while still keep updating $v_t^{(l)}$ in the future steps. Also stop updating $c_{avg}^{(l)}$.

4: **for** $t = T_w, \ldots, T$ **do**

5:     **(On $i$-th node)**

6:     Randomly sample $\xi_t^{(i)}$ and compute local stochastic gradient $g_t^{(i)} := \nabla F_i(x_t^{(i)}, \xi_t^{(i)})$, and update the local momentum $m_t^{(i)}$ according to $m_t^{(i)} = \beta_1 m_{t-1}^{(i)} + (1 - \beta_1)g_t^{(i)}$.

7:     Compress the fused momentum $m_t^{(i)}$ into $\hat{m}_t^{(i)} = C_\omega\left[m_t^{(i)} + \delta_{t-1}^{(i)}\right]$, and update the compression error by $\delta_t^{(i)} = m_t^{(i)} + \delta_{t-1}^{(i)} - \hat{m}_t^{(i)}$.

8:     Send the $\hat{m}_t^{(i)}$ to the server.

9:     **(On server)**

10:     Take the average over all $\hat{m}_t^{(i)}$ it receives and compress it into $\overline{m}_t = C_\omega\left[\frac{1}{n}\sum_{j=1}^n \hat{m}_t^{(i)} + \overline{\delta}_{t-1}\right]$, and update the compression error accordingly by $\overline{\delta}_t = \frac{1}{n}\sum_{j=1}^n \hat{m}_t^{(i)} + \overline{\delta}_{t-1} - \overline{m}_t$.

11:     Send $\overline{m}_t$ to all the workers.

12:     **(On $j$-th node)**

13:     Set $m_t = \overline{m}_t$.

14:     **for** the $l$-th layer **do**

15:         Reconstruct global gradient $g_t^{(l)} = (m_t^{(l)} - \beta_1 m_{t-1}^{(l)})/(1 - \beta_1)$.

16:         $v_t^{(l)} = \beta_2 v_{t-1}^{(l)} + (1 - \beta_2)\left(g_t^{(l)}\right)^2$.

17:         $r_t^{(l)} = \left\|v_{T_w}^{(l)}/v_t^{(l)}\right\|_\infty$.

18:         $r_t^{(l)} = \text{clip}\left(r_t^{(l)}, (1 - r_{threshold}) \times r_{t-1}^{(l)}, (1 + r_{threshold}) \times r_{t-1}^{(l)}\right)$.

19:         $r_t^{(l)} = \text{clip}\left(r_t^{(l)}, r_{min}, r_{max}\right)$.

20:         $c_t^{(l)} = r_t^{(l)} c_{avg}^{(l)}$.

21:         Update model of the $l$-th layer $x_t^{(l)} = x_{t-1}^{(l)} - \gamma c_t^{(l)}\frac{m_t^{(l)}}{\sqrt{v_{T_w}^{(l)}}}$.

22:     **end for**

23: **end for**

24: **Output**: $x$.

---

We summarize 1-bit LAMB in Algorithm 1. During the compression stage, we freeze the variance in order to apply the error compensation mechanism correctly (Tang et al., 2021). However, this brings two challenges for LAMB's case: 1) We cannot update LAMB's scaling coefficients ($c_t^{(l)}$ in (1)) during compression stage based on LAMB algorithm, because it requires uncompressed momentum and up-to-date variance; 2) In vanilla LAMB the scaling coefficients become stable during training. However, because LAMB's variance term is not stable for some layers but we freeze them during compression, we need to adjust scaling coefficients during compression to compensate this.

To this end, 1-bit LAMB uses a novel way to adaptively update LAMB scaling coefficients during the compression stage to compensate the difference between frozen and actual variance. During the warmup stage, we use vallia LAMB and keep track of the moving average of each layer's scaling coefficient (used during the compression stage because the scaling coefficient is not stable at beginning). At the end of warmup, we stop updating the moving average, and store the frozen variance to be used during compression stage. On the other hand, we still keep updating another "fresh" variance by reconstructing the global gradient based on this and last step's compressed momentum.

To update LAMB scaling coefficient during compression stage, we compute a 1-bit LAMB scaling ratio which is the max element among the (*frozen variance/fresh variance*). We use the max element as the scaling ratio because it is effective and cheap to compute based on our experiments. To avoid extreme ratios and dramatic change between ratios, we use two kinds of clipping configurable by the user. (The clipping from vanilla LAMB algorithm are not used during the compression stage.) Then we compute this step's LAMB scaling coefficient using the 1-bit LAMB scaling ratio and the moving average at the end of warmup, and use it to update the model. Our study shows that those layers with less stable variance tend to have more dynamic 1-bit LAMB scaling ratios, which demonstrate the desired adaptiveness (details in Appendix A.5). Appendix A.9 provides a theoretical analysis for 1-bit LAMB.

## 5 Proposed System Design

To realize the compressed communication at system level, Tang et al. (2021) designed a custom collective primitive, called "compressed allreduce", using Message Passing Interface (MPI). In addition to implementing 1-bit LAMB using this MPI-based backend, we introduce a new system implementation for compressed communication using the NCCL backend of PyTorch distributed, following the 3-phase design in the MPI-based backend: 1) The gather step, where each worker sends its $i$-th chunk to worker $i$, is implemented using NCCL's Alltoall and AllGather. 2) The average step, where each worker averages all chunks it receives. Here each worker acts as the server defined in Algorithm 1. 3) The scatter step, where each worker receives the average of all $i$-th chunks from worker $i$, is implemented using NCCL's AllGather. Compared to the MPI-based, this new NCCL-based implementation significantly improves the usability since NCCL is integrated with PyTorch distributed. In addition, evaluations show that the performance of the NCCL-based implementation is better than the MPI-based for Ethernet-based systems and on-par for InfiniBand-based systems. Thus we will mainly present the results with NCCL-based backend, but we also include a comparison between MPI and NCCL-based implementations in Appendix A.6.

## 6 Evaluation

**Dataset and models**   We evaluate the convergence and performance of 1-bit LAMB and uncompressed LAMB for BERT-Large ($L = 24$, $H = 1024$, $A = 16$, $340M$ params) pre-training task. We use the same dataset as Devlin et al. (2019), which is a concatenation of Wikipedia and BooksCorpus with 2.5B and 800M words respectively. Compared to the original BERT model, one notable change is that we applied PreLN instead of PostLN for better training stability (Zhang & He, 2020; Xiong et al., 2020). We use the GLUE fine-tuning benchmark (Wang et al., 2018) and SQuAD 1.1 fine-tuning task[4] to evaluate the convergence of the BERT models trained by LAMB and 1-bit LAMB.

**Hardware**   We use the two clusters described in Section 3.2. We use 8 to 256 GPUs for BERT pre-training tasks to measure 1-bit LAMB's performance gain. For fine-tuning tasks we use 4 GPUs. One thing to note is that because 1-bit LAMB introduces additional memory overhead (one persistent copy of the fresh varaince and one temporary copy of last step's momentum) and because the V100 GPUs on the Ethernet cluster have 16GB memory instead of 32GB, we were not able to fit large batch sizes for BERT pre-training when the number of GPUs is small. For seqlen 128 and 512, we need to use at least 8 and 16 GPUs on the Ethernet cluster, respectively.

**Training parameters**   For BERT pre-training, we set the parameters in (1) as $\beta_1 = 0.9$, $\beta_2 = 0.999$, $c_{min} = 0.01$ and $c_{max} = 0.3$ for LAMB and 1-bit LAMB. For 1-bit LAMB, we set $\beta_3 = 0.9$, $r_{min} = 0.5$, $r_{max} = 4.0$, and $r_{threshold} = 0.1$ in Algorithm 1. For convergence analysis, we set total batch size as 64K for seqlen 128 and 32K for seqlen 512. For performance analysis, we test different batch sizes from 8K to 64K.

For BERT pre-training seqlen 128, the learning rate starts from $1 \times 10^{-3}$, exponentially increases to $12 \times 10^{-3}$ as a warmup in the first 450 steps, then decays into 0.9 of the original after every 250 steps. The total number of steps is 5993. For 1-bit LAMB we use the first 1000 steps (16.7%) as the warmup stage. For BERT pre-training seqlen 512, the learning rate starts from 0, exponentially increases to $2 \times 10^{-3}$ as a warmup in the first 150 steps, then decays into 0.9 of the original after

---

[4]https://rajpurkar.github.io/SQuAD-explorer/

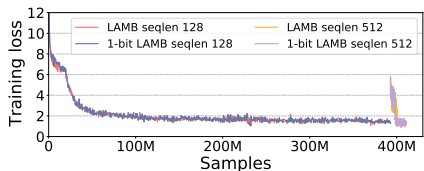

Figure 2: Sample-wise convergence speed for BERT-Large pre-training with LAMB and 1-bit LAMB.

Table 3: GLUE development set results using the pre-trained BERT-Large models. "Original" results are from Devlin et al. (2019) using BertAdam. "LAMB" results use the uncompressed LAMB for BERT pre-training. "LAMB + basic 1-bit" is the experimental algorithm described in Section 3.4. "1-bit LAMB" is the proposed work. The latter 3 cases use the same shared training parameters during pre-training and fine-tuning. Spearman correlations are reported for STS-B, and accuracy scores are reported for the other tasks. Each task's scores are the median scores over 32 runs.

|  | MNLI-(m/mm) | QQP | QNLI | SST-2 | CoLA | STS-B | MRPC | RTE | **Average** |
|---|---|---|---|---|---|---|---|---|---|
| Original | 86.7/85.9 | 89.3 | 92.7 | 94.9 | 60.5 | 86.5 | 85.4 | 70.1 | 83.6 |
| LAMB | 85.4/85.5 | 91.4 | 91.9 | 92.8 | 60.9 | 90.1 | 86.9 | 70.4 | **83.9** |
| LAMB + basic 1-bit | 84.8/84.7 | 91.2 | 91.4 | 92.4 | 54.0 | 89.6 | 84.2 | 66.6 | 82.1 |
| 1-bit LAMB | 85.5/85.6 | 91.3 | 92.3 | 93.1 | 59.2 | 90.0 | 86.5 | 71.5 | **83.9** |

every 150 steps. The total number of steps is 555. For 1-bit LAMB we use the first 107 steps (19.3%) as the warmup stage.

For 1-bit LAMB's hyperparameter tuning, we find that the only parameter that requires nontrivial tuning is the number of warmup steps, which depends on when do the LAMB's scaling coefficients become stable as discussed in Section 3.3 thus could potentially be automated. For the other parameters ($\beta_3$, $r_{min}$, $r_{max}$, and $r_{threshold}$ in Algorithm 1) we find tuning them does not lead to very different convergence since they are just used to avoid extreme value/dramatic change of LAMB's scaling coefficients. Similarly the clipping parameters $c_{min}$ and $c_{max}$ in (1) for LAMB and 1-bit LAMB do not require non-trivial tuning.

For GLUE benchmarks we use Adam optimizer and perform single-task training on the dev set. Following the setup in the BERT paper (Devlin et al., 2019), we use a batch size of 32 and fine-tune for 3 epochs for all GLUE tasks. For each task, we select the best learning rate among $\{2 \times 10^{-5}, 3 \times 10^{-5}, 4 \times 10^{-5}, 5 \times 10^{-5}\}$. For SQuAD fine-tuning we use Adam optimizer and the same parameters as published by HuggingFace (batch size = 24, learning rate = $3 \times 10^{-5}$, dropout = 0.1, 2 epochs).

**Convergence analysis** Figure 2 presents the BERT pre-training sample-wise convergence results. For both seqlen 128 and 512, 1-bit LAMB provides the same convergence speed as LAMB, while it takes much less time to process each batch due to communication compression. We already presented the BERT pre-training validation loss and SQuAD fine-tuning results in Section 3.4 Table 2. And Table 3 presents the GLUE results. For both SQuAD and GLUE, results show that 1-bit LAMB provides the same fine-tuning task accuracy as LAMB, while simply freezing both the varaince and LAMB's scaling coefficients would hurt the accuracy.

**Performance analysis** Computed as 1/(warmup_ratio + (1 - warmup_ratio)/16) for FP16 training, 1-bit LAMB offers $4.6\times$ and $4.1\times$ end-to-end communication volume reduction for BERT-Large seqlen 128 and 512, respectively. To measure the actual end-to-end performance gain, first we perform a throughput analysis where we run the warmup (i.e., baseline LAMB's performance) and compression stage of 1-bit LAMB for 200 steps each, and measure the average throughput of the two stages. Figure 3 presents the results with NCCL-based compressed communication backend under different batch sizes and number of GPUs. For seqlen 128, 1-bit LAMB provides up to $4.5\times$ speedup during the compression stage, which is equivalent to $2.8\times$ end-to-end speedup (computed as 1/(warmup_ratio + (1 - warmup_ratio)/compression_stage_speedup)). For seqlen 512, 1-bit LAMB provides up to $3.0\times$ speedup during the compression stage, which is equivalent to $2.2\times$ end-to-end speedup. This demonstrates 1-bit LAMB's better scalability compared to LAMB. It is also worth mentioning that 1-bit LAMB on Ethernet (4.1 Gbps effective bandwidth, 4 GPUs per node) is able

Table 4: Total runtime of BERT-Large pre-training with LAMB and 1-bit LAMB (256 GPUs with Ethernet connections. Batch sizes 64K/32K for seqlen 128/512.)

|  | Seqlen 128 | Seqlen 512 | Total |
|---|---|---|---|
| LAMB | 657 min | 74 min | 731 min |
| 1-bit LAMB | 301 min (2.2x) | 50 min (1.5x) | 351 min (2.1x) |

(a) seqlen 128, batch size 8K  (b) seqlen 128, batch size 16K  (c) seqlen 128, batch size 32K  (d) seqlen 128, batch size 64K

(e) seqlen 512, batch size 8K  (f) seqlen 512, batch size 16K  (g) seqlen 512, batch size 32K

Figure 3: Scalability of 1-bit LAMB with NCCL-based backend for BERT-Large pre-training on V100 GPUs. LAMB lines represent the throughput at 1-bit LAMB's warmup stage (i.e., baseline LAMB). 1-bit LAMB lines represent the throughput at compression stage. Annotations represent the highest speedup achieved in each figure. Note that this is the speedup between warmup and compression stage. The end-to-end speedup also depends on the percentage of warmup. All figures share the same legend as 3(g).

to achieve comparable throughput as LAMB on InfiniBand (near 100 Gbps effective bandwidth, 8 GPUs per node), which demonstrates 1-bit LAMB's efficiency considering the hardware differences.

In addition to throughput analysis, we also measure the total runtime of pre-training at batch size $64/32K$ for both 1-bit LAMB and LAMB. As shown in Table 4, overall 1-bit LAMB is able to provide $2.2\times$ and $1.5\times$ speedup for seqlen 128 and 512. These numbers are consistent with the end-to-end speedup calculated in the throughput analysis ($2.0\times$ and $1.5\times$ based on results in Figure 3(d) and 3(g)). For seqlen 128, the end-to-end speedup based on runtime is slightly higher than the speedup based on throughput. We find that it is because uncompressed LAMB's larger communication volume makes it more sensitive to the occasional fluctuation of the actual network bandwidth.

## 7 CONCLUSION

To reduce both the frequency and volume of communications for large-scale training, we propose an error-compensated LAMB preconditioned momentum SGD algorithm, 1-bit LAMB, which combines the power of large batch optimization and communication compression by introducing an novel way to support adaptive layerwise learning rates during communication compression. We also introduce an easier-to-use and more efficient compressed communication backend system based on NCCL. Evaluations show that 1-bit LAMB with NCCL-based backend is able to achieve up to $4.6\times$ communication volume reduction and up to $2.8\times$ end-to-end time-wise speedup for BERT pre-training compared to uncompressed LAMB, together with the same sample-wise convergence speed and fine-tuning task accuracy.

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

Table 5: BERT-Large pre-training seqlen 128 profiling results.

| Cluster Network Type | Num. node | Num. GPU | Batch size per GPU | Batch size | Grad accum. step | Forward (ms) | Backward allreduce (ms) | Backward everything else (ms) | Step (ms) | allreduce% |
|---|---|---|---|---|---|---|---|---|---|---|
| Ethernet | 64 | 256 | 16 | 8K | 2 | 55 | 3579 | 117 | 191 | **91%** |
| Ethernet | 64 | 256 | 16 | 16K | 4 | 111 | 3533 | 227 | 195 | 87% |
| Ethernet | 64 | 256 | 16 | 32K | 8 | 224 | 3599 | 462 | 233 | 80% |
| Ethernet | 64 | 256 | 16 | 64K | 16 | 445 | 3674 | 919 | 215 | 70% |
| Ethernet | 32 | 128 | 16 | 8K | 4 | 112 | 3759 | 233 | 121 | 89% |
| Ethernet | 16 | 64 | 16 | 8K | 8 | 223 | 3433 | 464 | 109 | 81% |
| Ethernet | 8 | 32 | 16 | 8K | 16 | 445 | 3528 | 923 | 38 | 72% |
| Ethernet | 4 | 16 | 16 | 8K | 32 | 881 | 3436 | 1827 | 33 | 56% |
| Ethernet | 2 | 8 | 16 | 8K | 64 | 1773 | 2087 | 3696 | 31 | 28% |
| Ethernet | 1 | 4 | 16 | 8K | 128 | 3532 | 234 | 7329 | 30 | 2% |
| InfiniBand | 16 | 128 | 64 | 8K | 1 | 96 | 335 | 179 | 36 | **52%** |
| InfiniBand | 16 | 128 | 64 | 16K | 2 | 192 | 346 | 356 | 37 | 37% |
| InfiniBand | 16 | 128 | 64 | 32K | 4 | 381 | 377 | 714 | 37 | 25% |
| InfiniBand | 16 | 128 | 64 | 64K | 8 | 770 | 422 | 1422 | 32 | 16% |
| InfiniBand | 8 | 64 | 64 | 8K | 2 | 192 | 332 | 352 | 34 | 36% |
| InfiniBand | 4 | 32 | 64 | 8K | 4 | 384 | 339 | 711 | 31 | 23% |
| InfiniBand | 2 | 16 | 64 | 8K | 8 | 768 | 270 | 1436 | 31 | 11% |
| InfiniBand | 1 | 8 | 64 | 8K | 16 | 1534 | 167 | 2869 | 31 | 4% |

## A    APPENDIX

### A.1    TRAINING PARAMETERS FOR 1-BIT ADAM EXPERIMENTS IN SECTION 3.1

For experiments in Section 3.1, for both LAMB and 1-bit Adam we use batch size = 16K, 28125/3125 steps for seqlen 128/512, weight decay = 0.01, linear LR warmup and decay. For LAMB, we use learning rate = $3.54 \times 10^{-3}$, 10% LR warmup, clipping configs ($c_{min}$ and $c_{max}$ in (1)) as 0.1 and 1. For 1-bit Adam, we use learning rates $\in \{1 \times 10^{-4}, 2 \times 10^{-4}, 3 \times 10^{-4}\}$, LR warmup $\in \{5\%, 10\%, 20\%\}$. All of these training parameters (except LAMB clipping configs) are from the LAMB paper. For 1-bit Adam, following the original work's strategy we set the number of warmup steps as 4000 (out of total 28125 steps) for seqlen 128 and 475 (out of 3125) for seqlen 512.

### A.2    DETAILED RESULTS FOR PROFILING EXPERIMENTS IN SECTION 3.2

Table 5 presents the detailed profiling results. Results show that even with larger batch sizes the allreduce communication still contributes to a great portion of the training time per step, up to 91% and 52% on two different kinds of clusters. And this overhead is larger when the number of nodes is larger, when the batch size is smaller, when the network bandwidth is lower.

### A.3    DETAILED RESULTS FOR LAMB EXPERIMENTS IN SECTION 3.3

Figure 4 presents LAMB's scaling coefficients for different layers during BERT pre-training sequence length 128 (sequence length 512 has similar patterns). Only the `cls.seq_relationship.bias` has a very unstable scaling coefficient. This is because this bias only has two elements, representing the two states of whether the two sequences are next to each other.

Figure 5 presents LAMB's variance norms for different layers during BERT pre-training sequence length 128 (sequence length 512 has similar patterns). We believe that there are two reasons why LAMB's varaince terms are less stable: 1) LAMB has larger batch size, smaller number of steps, and layerwise adaptive learning rates compared to Adam. 2) Because LAMB requires the calculation of the scaling coefficient for each layer, we cannot fuse all the variance together as in Adam. And each separate variance could have less stable norm compared to a single fused variance.

### A.4    DESIGN OF THE EXPERIMENTAL ALGORITHM IN SECTION 3.4

Because LAMB's scaling coefficient is essentially part of the learning rate (and it's just a scalar), up-dating this coefficient would not affect communication compression's error compensation mechanism

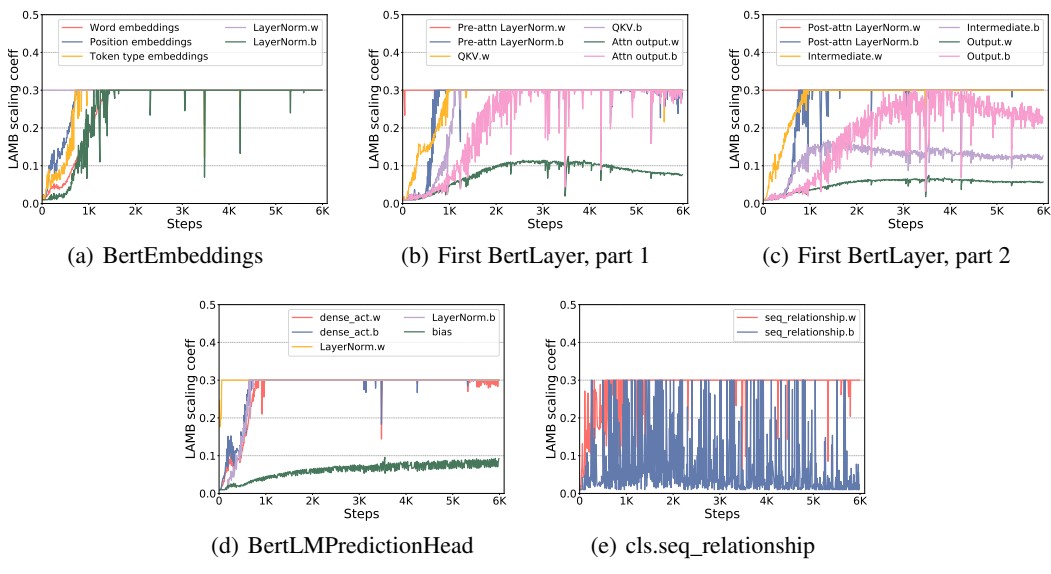

(a) BertEmbeddings  (b) First BertLayer, part 1  (c) First BertLayer, part 2

(d) BertLMPredictionHead  (e) cls.seq_relationship

Figure 4: LAMB's scaling coefficients ($c_t^{(l)}$ in (1)) for different layers during BERT-Large pre-training seqlen 128 (5993 steps in total). Since all 24 BertLayer have similar patterns, we just present the first one. We set the lower/upper bound of the scaling coefficient ($c_{min}$ and $c_{max}$ in (1)) at 0.01 and 0.3.

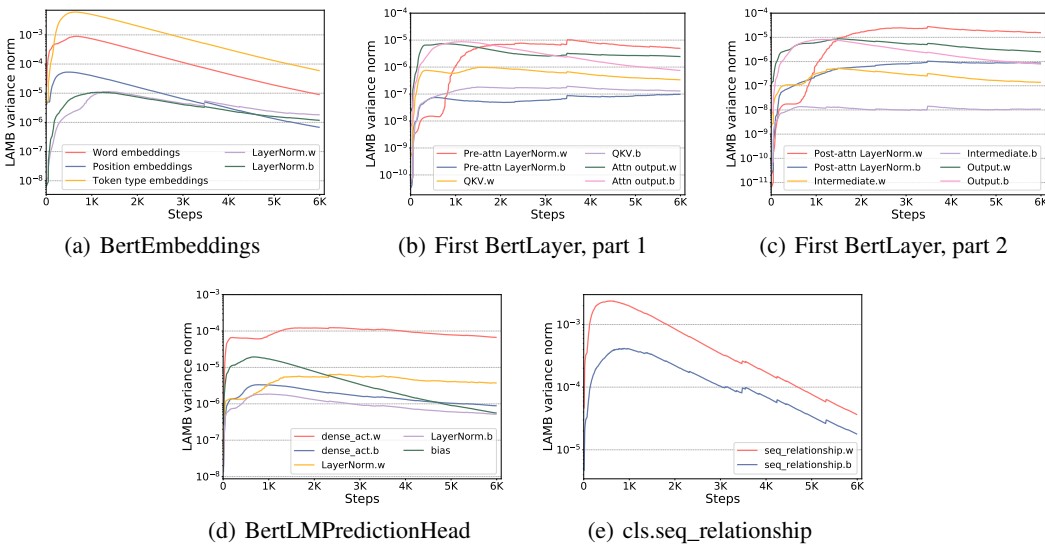

(a) BertEmbeddings  (b) First BertLayer, part 1  (c) First BertLayer, part 2

(d) BertLMPredictionHead  (e) cls.seq_relationship

Figure 5: LAMB's variance norms for different layers during BERT-Large pre-training seqlen 128. Since all 24 BertLayer have similar patterns, we just present the first one. The y-axis is in log scale.

as long as the change is small enough between two steps.[5] However, we find that it's challenging to update this coefficient during compression stage because in LAMB's algorithm, updating this scaling coefficient requires both momentum and variance term. However, the error compensation mechanism requires freezing the variance term at the beginning of the compression stage due to its nonlinear dependency to the gradient. In addition, we find that due to the error compensation mechanism, the norm of some layers' momentum term could become larger/smaller compared to the uncompressed case. As a result, we find that updating LAMB's scaling coefficients during compression stage based

---

[5]In fact as described in Section 4, the proposed 1-bit LAMB algorithm does require updating LAMB's scaling coefficient during compression stage, but in a way different from original LAMB algorithm.

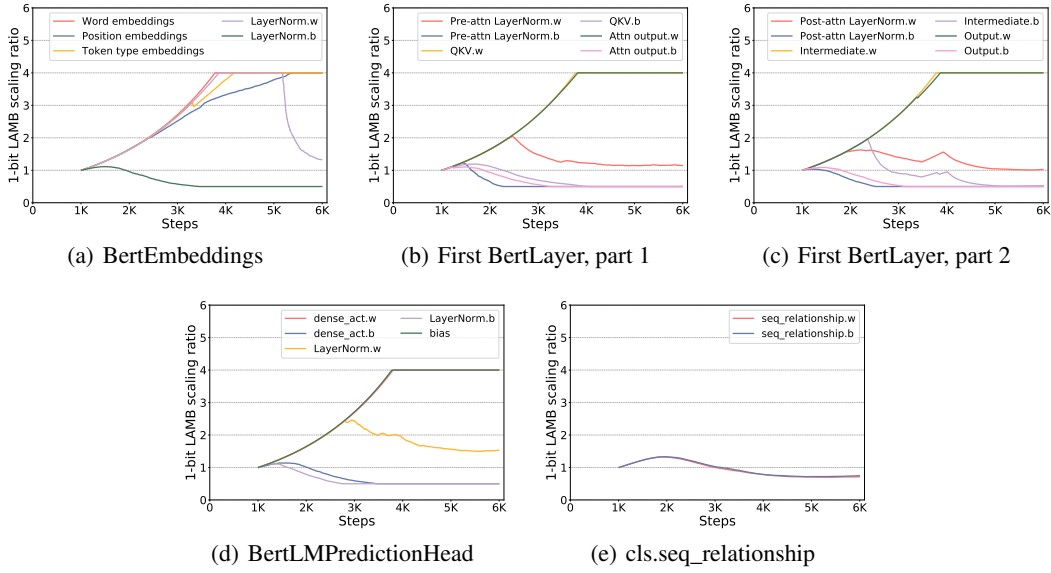

Figure 6: 1-bit LAMB scaling ratios ($r_t^{(l)}$ in Algorithm 1) for different layers during BERT pre-training sequence length 128. Since all 24 BertLayer have similar patterns, we just present the first one. The number of warmup steps is $1K$ out of 5993 total steps. We set the clipping configurations of the scaling ratio ($r_{min}, r_{max}, r_{threshold}$ in Algorithm 1) at 0.5, 4.0, 0.1.

on original LAMB algorithm would produce suboptimal results and slower convergence speed. Thus for Section 3.4's experimental algorithm, we stop updating LAMB's scaling coefficients during compression stage, and just use a calculated scaling coefficient moving average at the end of the warmup stage. For compression stage we also tried communicating based on 1-bit compressed gradient, but it leads to much slower convergence speed. We believe it's because gradients are less stable, which could lead to higher compression error and slower convergence.

## A.5 1-BIT LAMB ALGORITHM DESIGN AND IMPLEMENTATION DETAILS

Figure 6 presents 1-bit LAMB scaling ratios ($r_t^{(l)}$ in Algorithm 1) for different layers during BERT pre-training sequence length 128 (sequence length 512 has similar patterns). When comparing with Figure 5, we find that for those layers with less stable varaince (e.g., 3 kinds of embeddings, weights in BertLayer), the corresponding 1-bit LAMB scaling ratios are also larger. As a result 1-bit LAMB is able to adaptively update LAMB scaling coefficients during compression according to the difference between the frozen and fresh variance.

### A.5.1 REDUCING NUMBER OF COMMUNICATIONS BY MOMENTUM FUSION

Different from Adam, LAMB has the scaling coefficients that need to be updated separately for each layer. For the communication of the compressed momentum during 1-bit LAMB's compression stage, if we also communicate separately for each layer, the number of communications (which is 302 for BERT-Large model) will greatly affect the overall performance. Thus we fuse all layers' momentum into a contiguous 1D buffer and just do one communication over this fused momentum. This fused momentum is not a duplication, but a different "view" of all layers' momentum. We implement this momentum fusion by `torch._utils._flatten_dense_tensors` and `torch._utils._unflatten_dense_tensors`.

### A.5.2 REDUCING COMPRESSION ERROR BY MOMENTUM SCALING

We find that after momentum fusion, the compression error increase a lot for some layers which greatly increase the chance of divergence. This is because: 1) When communicating base on the fused momentum, all layers' momentum will be compressed to the same scale due to the nature of

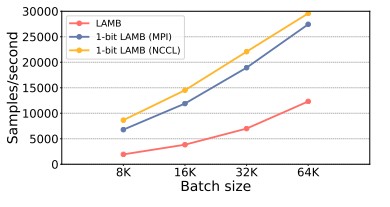 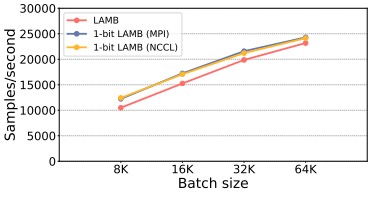

(a) 256 GPUs on Ethernet cluster  (b) 128 GPUs on InfiniBand cluster

Figure 7: Comparing MPI and NCCL-based compressed communication backend implementations based on performance of BERT pre-training seqlen 128.

1-bit compression (representing the tensor by $\pm 1$ signs and a *single* scale). Thus the layers with very small/large momentum scale will have larger compression error; 2) Due to LAMB's layerwise adaptive learning rate, each layer is learning at different speed, which could further increase the variance of momentum scale among layers. We solve this issue by computing an average momentum scale at the end of warmup stage and using it to compute a momentum scale coefficient for each layer. During the compression stage, we multiply each layer's local momentum by its scale coefficient, then do the compressed communication, then divide each layer's global momentum by the same scale coefficient. By performing this momentum scaling, all layers' momentum will have similar scales when passed to the compressed communication. Thus we are able to greatly reduce the compression error and chance of divergence. Since the momentum scale coefficients are fixed during the compression stage, it won't affect 1-bit compression's error compensation mechanism.

### A.6    COMPARING MPI AND NCCL-BASED COMMUNICATION BACKEND

All of the results in main paper are evaluated with the NCCL-based compressed communication backend implementation. Figure 7 presents the performance comparison between MPI and NCCL-based implementations. Compared to the MPI-based implementation introduced in the 1-bit Adam work, our NCCL-based implementation is able to provide better performance on the Ethernet cluster (where OpenMPI library is used for MPI backend) and on par performance on the InfiniBand cluster (where MVAPICH2-GDR is used for MPI backend).

### A.7    ADDITIONAL CONVERGENCE ANALYSIS ABOUT NUMBER OF WORKERS AND BATCH SIZE

This section provides additional convergence analysis on BERT pre-training and SQuAD fine-tuning. First we investigate whether the number of workers/GPUs affect the convergence of 1-bit LAMB. Table 6 presents the SQuAD average/max F1 scores using the BERT models pre-trained with LAMB and 1-bit LAMB optimizers, with different number of GPUs, and with the same training parameters described in Section 6. Results show that both LAMB and 1-bit LAMB achieve similar SQuAD F1 scores under different number of workers, and 1-bit LAMB always provides better results.

Next, we investigate whether the batch size affects the convergence of 1-bit LAMB. In main paper we compared LAMB and 1-bit LAMB when pre-training BERT with batch size 64K/32K at seqlen 128/512. Here we compare them at batch size 16K and 4K. At batch size 16K, for LAMB we use the same training parameters described in Appendix A.1, and for 1-bit LAMB we set the number of warmup steps as 4000 (14%) for seqlen 128 and 475 (15%) for seqlen 512. At batch size 4K, for LAMB we follow the original work and use 112500/12500 steps for seqlen 128/512, weight decay = 0.01, linear LR warmup and decay, learning rate = $1.77 \times 10^{-3}$, 2.5% LR warmup, clipping configs ($c_{min}$ and $c_{max}$ in (1)) as 0.1 and 1. For 1-bit LAMB we set the number of warmup steps as 16000 (14%) for seqlen 128 and 1900 (15%) for seqlen 512. For 1-bit LAMB's other hyperparameters in Algorithm 1, we keep them the same under all batch sizes: $\beta_3 = 0.9$, $r_{min} = 0.5$, $r_{max} = 4.0$, and $r_{threshold} = 0.1$. As shown in Table 7, results show that 1-bit LAMB provides better SQuAD F1 scores than LAMB under all batch sizes.

Table 6: SQuAD average/max dev set F1 scores over 32 runs using the BERT models pre-trained (batch size 64K/32K at seqlen 128/512) with different optimizers and different number of GPUs/workers. The first column is from the original LAMB work. The last two columns are our experiments using the same training parameters.

| SQuAD Avg./Max F1 | LAMB (You et al., 2020) | LAMB | 1-bit LAMB |
|---|---|---|---|
| 128 GPUs | –/90.584 | 90.257/90.557 | 90.494/90.740 |
| 64 GPUs (from Table 2) | –/90.584 | 90.265/90.555 | 90.524/90.788 |
| 32 GPUs | –/90.584 | 90.263/90.584 | 90.469/90.811 |

Table 7: SQuAD average/max dev set F1 scores over 32 runs using the BERT models pre-trained with different optimizers and different pre-training batch sizes. The first column is from the original LAMB work. The last two columns are our experiments using the same training parameters.

| SQuAD Avg./Max F1 | LAMB (You et al., 2020) | LAMB | 1-bit LAMB |
|---|---|---|---|
| Bsz 64K/32K for seqlen 128/512 (from Table 2) | –/90.584 | 90.265/90.555 | 90.524/90.788 |
| Bsz 16K/16K for seqlen 128/512 | –/91.345 | 90.716/91.119 | 90.906/91.142 |
| Bsz 4K/4K for seqlen 128/512 | –/91.137 | 90.917/91.084 | 90.942/91.171 |

## A.8   ADDITIONAL CONVERGENCE ANALYSIS: RESNET-50 ON CIFAR100

This section provides additional analysis of training ResNet-50 on CIFAR100. Because ResNet-50 is much smaller than BERT-Large and because we use 8 GPUs on a single node for the training, 1-bit LAMB and LAMB have very similar throughput. Thus this experiment focus on whether 1-bit LAMB can achieve the same convergence as LAMB under the same training hyperparameters. For both LAMB and 1-bit LAMB, we use 250 epochs (about 12K steps), batch size 1K, learning rates $\in \{0.005, 0.01, 0.02\}$, linear LR warmup in the first 1K steps, no LR decay, weight decay = 0.01. For LAMB we set the clipping configs ($c_{min}$ and $c_{max}$ in (1)) as 0.01 and 0.3. For 1-bit LAMB we set the number of warmup steps as 1K (8%), and set other hyperparameters in Algorithm 1 as: $\beta_3 = 0.9$, $r_{min} = 0.5$, $r_{max} = 2.0$, and $r_{threshold} = 0.1$. Figure 8 presents the sample-wise training loss and testing accuracy, and Table 8 presents the final testing accuracy. Results show that under all three learning rates, 1-bit LAMB provides on par or better convergence and final accuracy than LAMB.

## A.9   THEORETICAL ANALYSIS

Notice that for 1-bit LAMB, we only use original LAMB at warm-up, and then we essentially run error-compensated momentum SGD with coordinate-dependent learning rate $\frac{\gamma}{\sqrt{v_{T_w}}}$. Therefore here we consider the LAMB-based warm-up phase as a way to find a good precondition variance term $v_{T_w}$ to be used in the compression phase. Below we are going to introduce the convergence rate for the compression phase after warm-up. We first introduce some necessary assumptions, then we present the theoretical guarantee of the convergence rate for 1-bit LAMB.

**Assumption 1.** *We make the following assumptions:*

1. ***Lipschitzian gradient:*** *$f(\cdot)$ is assumed to be with L-Lipschitzian gradients, which means*
$$\|\nabla f(\boldsymbol{x}) - \nabla f(\boldsymbol{y})\| \leq L\|\boldsymbol{x} - \boldsymbol{y}\|, \quad \forall \boldsymbol{x}, \forall \boldsymbol{y},$$

2. ***Bounded variance:*** *The variance of the stochastic gradient is bounded*
$$\mathbb{E}_{\boldsymbol{\zeta}^{(i)} \sim \mathcal{D}_i} \|\nabla F(\boldsymbol{x}; \boldsymbol{\zeta}^{(i)}) - \nabla f(\boldsymbol{x})\|^2 \leq \sigma^2, \quad \forall \boldsymbol{x}, \forall i.$$

3. ***Bounded magnitude of error for $\mathcal{C}_\omega[\cdot]$:*** *The magnitude of worker's local errors $\boldsymbol{\delta}_t^{(i)}$ and the server's global error $\overline{\boldsymbol{\delta}}_t$, are assumed to be bounded by a constant $\epsilon$*
$$\sum_{k=1}^{n} \mathbb{E}_\omega \left\| \boldsymbol{\delta}_t^{(i)} \right\| \leq \frac{\epsilon}{2}, \quad \sum_{i=1}^{n} \mathbb{E}_\omega \left\| \overline{\boldsymbol{\delta}}_t \right\| \leq \frac{\epsilon}{2}, \quad \forall t, \forall i.$$

The most important reason we use the third assumption above is that for 1-bit compression, the signal-to-noise assumption does not hold if there is only one non-zero element in the tensor. In (Tang

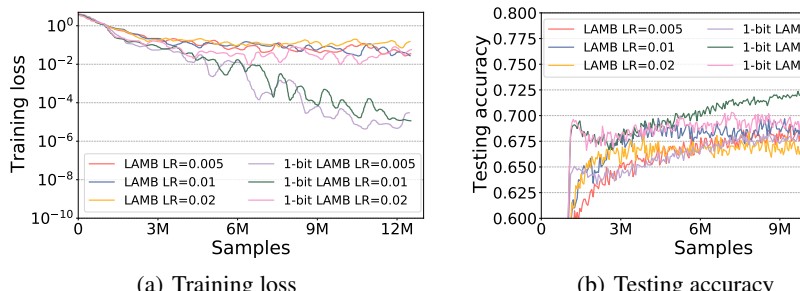

(a) Training loss

(b) Testing accuracy

Figure 8: Sample-wise training loss and testing accuracy for ResNet-50 on CIFAR100.

Table 8: Final testing accuracy for ResNet-50 on CIFAR100.

| Final testing accuracy | LAMB | 1-bit LAMB |
|---|---|---|
| LR=0.005 | 0.68 | 0.68 |
| LR=0.01 | 0.68 | 0.73 |
| LR=0.02 | 0.66 | 0.69 |

et al., 2019), authors proved that our assumption is equivalent to the signal-to-noise assumption when the gradient magnitude is upper bounded.

Next we present the main theorem for 1-bit LAMB.

**Theorem 1.** *Under Assumption 1, for 1-bit LAMB, we have the following convergence rate*

$$\left(1 - \gamma_{\max}LV_{\max} - \frac{8\gamma_{\max}^3 L^2}{(1-\beta)^2\gamma_{\min}}\right)\sum_{t=0}^{T}\mathbb{E}\|\nabla f(\boldsymbol{x}_t)\|^2$$
$$\leq \frac{2\mathbb{E}f(\boldsymbol{x}_1) - 2\mathbb{E}f(\boldsymbol{y}^*)}{\gamma_{\min}} + \frac{24\gamma_{\max}^3 L^2 V_{\max}^3 \epsilon^2 T}{(1-\beta)^2\gamma_{\min}} + \frac{LV_{\max}\gamma_{\max}^2\sigma^2 T}{n\gamma_{\min}} + \frac{8\gamma_{\max}^3 L^2 V_{\max}^2\sigma^2 T}{n(1-\beta)^2\gamma_{\min}}, \quad (3)$$

*where $V = diag\left(1/\boldsymbol{v}_{T_w}^{(1)}, 1/\boldsymbol{v}_{T_w}^{(2)}, \cdots, 1/\boldsymbol{v}_{T_w}^{(d)}\right)$ is a diagonal matrix spanned by $\boldsymbol{v}_{T_w}$.*

Here $1/\boldsymbol{v}$ is a moving sum over all history $(\nabla f(\boldsymbol{x}))^2$, not $\|\nabla f(\boldsymbol{x})\|$. Therefore it will not approach to $0$ for the following two reasons: i) We stop updating $\boldsymbol{v}$ after the warmup stage, which means $\nabla f(\boldsymbol{x})$ at this time will not approach $0$ since it has not converged yet; ii) Since $\beta_2 = 0.999$, this means $\boldsymbol{v}$ is heavily dependent to the history gradients, therefore even though $\nabla f(\boldsymbol{x})$ approaches to $0$, $\boldsymbol{v}$ itself will still taking a time to decay to $0$. Therefore it is reasonable for us to assume an upper bound for $V_{\max}$

Given the generic result in Theorem 1, we obtain the convergence rate for 1-bit LAMB with appropriately chosen learning rate $\gamma$.

**Corollary 1.** *Under Assumption 1, for 1-bit LAMB, choosing $\gamma = \gamma_{\max} = \gamma_{\min} = \frac{1-\beta}{16LV_{\max}+\sigma\sqrt{\frac{T}{n}}+T^{\frac{1}{3}}\epsilon^{\frac{2}{3}}}$, we have the following convergence rate*

$$\frac{V_{\max}}{T}\sum_{t=0}^{T-1}\mathbb{E}\|\nabla f(\boldsymbol{x}_t)\|_V^2 \lesssim \frac{\sigma}{\sqrt{nT}} + \frac{\epsilon^{\frac{2}{3}}}{T^{\frac{2}{3}}} + \frac{1}{T},$$

*where we treat $f(\boldsymbol{x}_1) - f^*$, $\beta$ and $L$ as constants.*

This result suggests that: 1-bit LAMB essentially admits the same convergence rate as distributed SGD in the sense that both of them admit the asymptotical convergence rate $O(1/\sqrt{nT})$, which means we can still achieve linear speedup w.r.t. the number of workers $n$.

For theoretical analysis, we follow the framework of previous work (Tang et al., 2021). Below we first summarize the updating rule of 1-bit LAMB, followed by the proof of Theorem 1, then the proof of Corollary 1.

### A.9.1 PROOF TO THE UPDATING FORM

Since our algorithm is equivalent to running a parameter-server prototype communication on each chunk of the gradient, so below we will assume a parameter-server model (which means the tensor is not required to be divided into $n$ chunks) for simplicity. Although we use a layer-wise updating strategy in 1-bit LAMB, for the simplicity of theoretical analysis, we consider the case where we use the same updating rule for all the layer, but our results can be easily applied to this layer-wise case.

According to the algorithm description in Algorithm 1, at iteration $t + 1$, the updating step of the momentum term $\boldsymbol{m}_{t+1}$ can be divided into two steps:

1. Local Update and Compress: each worker locally update $\boldsymbol{m}_t$ and use the error-compensate strategy for compressing.

$$\boldsymbol{m}_t^{(i)} = \beta \boldsymbol{m}_t + (1 - \beta) \boldsymbol{g}_t^{(i)}$$

$$\boldsymbol{m}_{t+\frac{1}{2}}^{(i)} = \mathcal{C}_\omega \left[ \boldsymbol{m}_t^{(i)} + \frac{c_{t-2}}{c_{t-1}} \boldsymbol{\delta}_t^{(i)} \right]$$

$$\boldsymbol{\delta}_{t+1}^{(i)} = \boldsymbol{m}_t^{(i)} + \boldsymbol{\delta}_t^{(i)} - \boldsymbol{m}_{t+\frac{1}{2}}^{(i)}.$$

2. All workers send its $\boldsymbol{m}_{t+\frac{1}{2}}^{(i)}$ to the server. The server takes the average over them and compress it again using error-compensation.

$$\boldsymbol{m}_{t+\frac{1}{2}} = \frac{1}{n} \sum_{i=1}^{n} \boldsymbol{m}_{t+\frac{1}{2}}^{(i)}$$

$$\boldsymbol{m}_{t+1} = \mathcal{C}_\omega \left[ \boldsymbol{m}_{t+\frac{1}{2}} + \frac{c_{t-2}}{c_{t-1}} \boldsymbol{\delta}_t \right]$$

$$\boldsymbol{\delta}_{t+1} = \boldsymbol{m}_{t+\frac{1}{2}} + \boldsymbol{\delta}_t - \boldsymbol{m}_{t+1}.$$

3. The server broadcast $\boldsymbol{m}_{t+1}$ to all workers, and all workers update the local model according to

$$\boldsymbol{x}_{t+1} = \boldsymbol{x}_t - \gamma c_{t-1} \frac{\boldsymbol{m}_{t+1}}{\sqrt{\boldsymbol{v}_{T_w}^2}}.$$

Notice that here we do not include the updating rule for $\{c_t\}$ because for the following part we will only view it as a scaling factor.

So actually the updating rule above can be summarized as

$$
\begin{aligned}
\boldsymbol{m}_{t+1} =& \mathcal{C}_\omega \left[ \boldsymbol{m}_{t+\frac{1}{2}} + \frac{c_{t-2}}{c_{t-1}} \boldsymbol{\delta}_t \right] \\
=& \boldsymbol{m}_{t+\frac{1}{2}} + \frac{c_{t-2}}{c_{t-1}} \boldsymbol{\delta}_t - \boldsymbol{\delta}_{t+1} \quad \text{(from the definition of } \boldsymbol{\delta}_{t+1}) \\
=& \frac{1}{n} \sum_{i=1}^{n} \mathcal{C}_\omega \left[ \boldsymbol{m}_t^{(i)} + \frac{c_{t-2}}{c_{t-1}} \boldsymbol{\delta}_t^{(i)} \right] + \frac{c_{t-2}}{c_{t-1}} \boldsymbol{\delta}_t - \boldsymbol{\delta}_{t+1} \\
=& \frac{1}{n} \sum_{i=1}^{n} \left( \boldsymbol{m}_t^{(i)} + \frac{c_{t-2}}{c_{t-1}} \boldsymbol{\delta}_t^{(i)} - \boldsymbol{\delta}_{t+1}^{(i)} \right) + \frac{c_{t-2}}{c_{t-1}} \boldsymbol{\delta}_t - \boldsymbol{\delta}_{t+1} \quad \text{(from the definition of } \boldsymbol{\delta}_{t+1}^{(i)}) \\
=& \beta \boldsymbol{m}_t + \frac{1 - \beta}{n} \sum_{i=1}^{n} \boldsymbol{g}_t^{(i)} + \frac{c_{t-2}}{c_{t-1}} \left( \frac{1}{n} \sum_{i=1}^{n} \boldsymbol{\delta}_t^{(i)} + \boldsymbol{\delta}_t \right) - \left( \frac{1}{n} \sum_{i=1}^{n} \boldsymbol{\delta}_{t+1}^{(i)} + \boldsymbol{\delta}_{t+1} \right).
\end{aligned}
$$

Denote

$$\bar{\boldsymbol{g}}_t = \frac{1}{n} \sum_{i=1}^{n} \boldsymbol{g}_t^{(i)}$$

$$\bar{\boldsymbol{\delta}}_t = \frac{1}{n} \sum_{i=1}^{n} \boldsymbol{\delta}_t^{(i)} + \boldsymbol{\delta}_t,$$

the update rule of $m_t$ can be summarized as

$$m_t = \beta m_{t-1} + (1 - \beta)\overline{g}_t + \frac{c_{t-2}}{c_{t-1}}\overline{\delta}_{t-1} - \overline{\delta}_t,$$

and

$$x_{t+1} = x_t - \gamma V c_{t-1} m_t,$$

where $V = \text{diag}(1/\sqrt{v_1}, 1/\sqrt{v_2}, \cdots, 1/\sqrt{v_d})$ is the a diagonal matrix that spanned with $v_{T_w}$. In order to simlify the notation, we define $\gamma_t := \gamma c_{t-1}$ as a time-varying learning rate, which reduces the updating rule above into

$$m_t = \beta m_{t-1} + (1 - \beta)\overline{g}_t + \frac{\gamma_{t-1}}{\gamma_t}\overline{\delta}_{t-1} - \overline{\delta}_t,$$

$$x_{t+1} = x_t - \gamma_t V m_t.$$

### A.9.2 PROOF TO THEOREM 1

Notice that in for 1-bit LAMB, the learning rate for each coordinate is different. In order to simplify our analysis, we instead consider another function that is defined as

$$H(z) = F(V^{\frac{1}{2}}z),$$

also

$$h(z) = f(V^{\frac{1}{2}}z),$$

where $V$ is a diagonal matrix.

In this case we have

$$V^{\frac{1}{2}}\nabla f(V^{\frac{1}{2}}z) = \nabla h(z).$$

Therefore the updating rule of 1-bit LAMB in the view of $h(\cdot)$ is

$$V^{\frac{1}{2}}z_{t+1} = V^{\frac{1}{2}}z_t - \gamma V^{\frac{1}{2}}\left(V^{\frac{1}{2}}m_t\right).$$

It can be easily verified that

$$m_t = (1 - \beta)\sum_{s=0}^{t}\beta^{t-s}\overline{g}_s + \sum_{s=0}^{t}\beta^{t-s}(\overline{\delta}_{s-1} - \overline{\delta}_s)$$

$$= (1 - \beta)\sum_{s=0}^{t}\beta^{t-s}\frac{1}{n}\sum_{i=1}^{n}\nabla F(V^{\frac{1}{2}}z_t; \xi_t^{(i)}) + \sum_{s=0}^{t}\beta^{t-s}(\overline{\delta}_{s-1} - \overline{\delta}_s)$$

which means

$$V^{\frac{1}{2}}m_t = (1 - \beta)\sum_{s=0}^{t}\beta^{t-s}\frac{1}{n}\sum_{i=1}^{n}V^{\frac{1}{2}}\nabla F(V^{\frac{1}{2}}z_t; \xi_t^{(i)}) + \sum_{s=0}^{t}\beta^{t-s}V^{\frac{1}{2}}(\overline{\delta}_{s-1} - \overline{\delta}_s)$$

$$= (1 - \beta)\sum_{s=0}^{t}\beta^{t-s}\frac{1}{n}\sum_{i=1}^{n}\nabla H(V^{\frac{1}{2}}z_t; \xi_t^{(i)}) + \sum_{s=0}^{t}\beta^{t-s}V^{\frac{1}{2}}(\overline{\delta}_{s-1} - \overline{\delta}_s)$$

$$= (1 - \beta)\sum_{s=0}^{t}\beta^{t-s}\overline{g}_s(z) + \sum_{s=0}^{t}\beta^{t-s}V^{\frac{1}{2}}(\overline{\delta}_{s-1} - \overline{\delta}_s),$$

where $\overline{g}_s(z)$ is the corresponding averaged stochastic gradient computed in the view of loss function $h(\cdot)$.

Then, if we define $m_t(z) = V^{\frac{1}{2}}m_t$, the updating rule of $m_t(z)$ admits

$$m_t(z) = \beta m_{t-1}(z) + (1 - \beta)\overline{g}_t(z) + V^{\frac{1}{2}}\overline{\delta}_{t-1} - V^{\frac{1}{2}}\overline{\delta}_t, \tag{4}$$

and

$$V^{\frac{1}{2}}\boldsymbol{z}_{t+1} = V^{\frac{1}{2}}\boldsymbol{z}_t - \gamma_t V^{\frac{1}{2}}\boldsymbol{m}_t(\boldsymbol{z})$$
$$\boldsymbol{z}_{t+1} = \boldsymbol{z}_t - \gamma_t \boldsymbol{m}_t(\boldsymbol{z}). \tag{5}$$

From (4) and (5) we shall see that using different learning rate for each coordinate is equivalent to optimizing a new loss function defined on scaling the original coordinate and using a uniform learning for all coordinates. Therefore below we first study the behavior of the error-compensated momentum SGD using a coordinate-independent time-varying learning rate.

Below are some critical lemmas for the proof of Theorem 1.

**Lemma 1.** *Given two non-negative sequences $\{a_t\}_{t=1}^{\infty}$ and $\{b_t\}_{t=1}^{\infty}$ that satisfying*

$$a_t = \sum_{s=1}^{t} \rho^{t-s} b_s, \tag{6}$$

*with $\rho \in [0, 1)$, we have*

$$D_k := \sum_{t=1}^{k} a_t^2 \leq \frac{1}{(1-\rho)^2} \sum_{s=1}^{k} b_s^2.$$

*Proof.* From the definition, we have

$$S_k = \sum_{t=1}^{k}\sum_{s=1}^{t} \rho^{t-s} b_s = \sum_{s=1}^{k}\sum_{t=s}^{k} \rho^{t-s} b_s = \sum_{s=1}^{k}\sum_{t=0}^{k-s} \rho^{t} b_s \leq \sum_{s=1}^{k} \frac{b_s}{1-\rho}, \tag{7}$$

$$D_k = \sum_{t=1}^{k}\sum_{s=1}^{t} \rho^{t-s} b_s \sum_{r=1}^{t} \rho^{t-r} b_r$$

$$= \sum_{t=1}^{k}\sum_{s=1}^{t}\sum_{r=1}^{t} \rho^{2t-s-r} b_s b_r$$

$$\leq \sum_{t=1}^{k}\sum_{s=1}^{t}\sum_{r=1}^{t} \rho^{2t-s-r} \frac{b_s^2 + b_r^2}{2}$$

$$= \sum_{t=1}^{k}\sum_{s=1}^{t}\sum_{r=1}^{t} \rho^{2t-s-r} b_s^2$$

$$\leq \frac{1}{1-\rho} \sum_{t=1}^{k}\sum_{s=1}^{t} \rho^{t-s} b_s^2$$

$$\leq \frac{1}{(1-\rho)^2} \sum_{s=1}^{k} b_s^2, \quad \text{(due to (7))}$$

which completes the proof. □

**Lemma 2.** *Under Assumption 1, for any sequence that follows the updating rule of*

$$\boldsymbol{x}_{t+1} = \boldsymbol{x}_t - \gamma_t \boldsymbol{m}_t$$
$$\boldsymbol{m}_t = \beta \boldsymbol{m}_{t-1} + (1-\beta)\overline{\boldsymbol{g}}_t + \frac{\gamma_{t-1}}{\gamma_t}\overline{\boldsymbol{\delta}}_{t-1} - \overline{\boldsymbol{\delta}}_t,$$

*if*

$$\mathbb{E}\overline{\boldsymbol{g}}_t = \nabla f(\boldsymbol{x}_t), \quad \mathbb{E}\|\overline{\boldsymbol{g}}_t - \nabla f(\boldsymbol{x}_t)\|^2 \leq \frac{\sigma^2}{n}, \quad \mathbb{E}\|\overline{\boldsymbol{\delta}}_t\|^2 \leq \epsilon^2, \quad \forall t,$$
$$\|\nabla f(\boldsymbol{x}) - \nabla f(\boldsymbol{y})\| \leq L\|\boldsymbol{x} - \boldsymbol{y}\|, \quad \forall \boldsymbol{x}, \forall \boldsymbol{y},$$

*then we can guarantee that*

$$\left(1 - \gamma L - \frac{2\gamma^2 L^2}{(1-\beta)^2}\right) \sum_{t=0}^{T} \mathbb{E}\|\nabla f(\boldsymbol{x}_t)\|^2$$

$$\leq \frac{2\mathbb{E}f(\boldsymbol{x}_1) - 2\mathbb{E}f(\boldsymbol{x}^*)}{\gamma} + \frac{6\gamma^2 L^2 \epsilon^2 T}{(1-\beta)^2} + \frac{L\gamma\sigma^2 T}{n} + \frac{2\gamma^2 L^2 \sigma^2 T}{n(1-\beta)^2}$$

*Proof.* Instead of investigating $\boldsymbol{x}_t$ directly, we introduce the following sequence

$$\boldsymbol{y}_t = \boldsymbol{x}_t - \frac{\gamma_t \boldsymbol{m}_t + \gamma_{t-1}\overline{\boldsymbol{\delta}}_t}{1-\beta}.$$

The updating rule of $\boldsymbol{y}_t$ admits

$$\boldsymbol{y}_{t+1} - \boldsymbol{y}_t = \boldsymbol{x}_{t+1} - \boldsymbol{x}_t - \frac{\gamma_t}{1-\beta}(\boldsymbol{m}_{t+1} - \boldsymbol{m}_t) + \frac{\gamma_t \overline{\boldsymbol{\delta}}_{t+1} - \gamma_{t-1}\overline{\boldsymbol{\delta}}_t}{1-\beta}$$

$$= -\gamma_t \boldsymbol{m}_t - \frac{\gamma_t}{1-\beta}\left(\beta \boldsymbol{m}_t + (1-\beta)\boldsymbol{g}_{t+1} - \overline{\boldsymbol{\delta}}_{t+1} + \frac{\gamma_{t-1}}{\gamma_t}\overline{\boldsymbol{\delta}}_t\right) + \frac{\gamma_t \overline{\boldsymbol{\delta}}_{t+1} - \gamma_{t-1}\overline{\boldsymbol{\delta}}_t}{1-\beta}$$

$$= -\gamma_t \boldsymbol{g}_{t+1}.$$

Since $f(\cdot)$ is with L-Lipschitzian, we have

$$\mathbb{E}f(\boldsymbol{y}_{t+1}) - \mathbb{E}f(\boldsymbol{y}_t)$$

$$\leq \mathbb{E}\langle \nabla f(\boldsymbol{y}_t), \boldsymbol{y}_{t+1} - \boldsymbol{y}_t\rangle + \frac{L}{2}\mathbb{E}\|\boldsymbol{y}_{t+1} - \boldsymbol{y}_t\|^2$$

$$= -\gamma_t \mathbb{E}\langle \nabla f(\boldsymbol{y}_t), \boldsymbol{g}_{t+1}\rangle + \frac{L\gamma_t^2}{2}\mathbb{E}\|\boldsymbol{g}_{t+1}\|^2$$

$$= -\gamma_t \mathbb{E}\langle \nabla f(\boldsymbol{y}_t), \nabla f(\boldsymbol{x}_{t+1})\rangle + \frac{L\gamma_t^2}{2}\mathbb{E}\|\boldsymbol{g}_{t+1}\|^2$$

$$= -\frac{\gamma_t}{2}\mathbb{E}\|\nabla f(\boldsymbol{x}_{t+1})\|^2 - \frac{\gamma_t}{2}\mathbb{E}\|\nabla f(\boldsymbol{y}_t)\|^2 + \frac{\gamma_t}{2}\mathbb{E}\|\nabla f(\boldsymbol{x}_{t+1}) - \nabla f(\boldsymbol{y}_t)\|^2 + \frac{L\gamma_t^2}{2}\mathbb{E}\|\boldsymbol{g}_{t+1}\|^2$$

$$\leq -\frac{\gamma_t}{2}\mathbb{E}\|\nabla f(\boldsymbol{x}_{t+1})\|^2 + \frac{L\gamma_t}{2}\mathbb{E}\|\boldsymbol{x}_{t+1} - \boldsymbol{y}_t\|^2 + \frac{L\gamma_t^2}{2}\mathbb{E}\|\boldsymbol{g}_{t+1}\|^2$$

$$= -\frac{\gamma_t}{2}\mathbb{E}\|\nabla f(\boldsymbol{x}_{t+1})\|^2 + \frac{\gamma_t^3 L^2}{2}\mathbb{E}\left\|\frac{(2-\beta)\boldsymbol{m}_t}{1-\beta} + \frac{\frac{\gamma_{t-1}}{\gamma_t}\overline{\boldsymbol{\delta}}_{t-1}}{1-\beta}\right\|^2 + \frac{L\gamma_t^2}{2}\mathbb{E}\|\boldsymbol{g}_{t+1}\|^2$$

$$\leq -\frac{\gamma_t}{2}\mathbb{E}\|\nabla f(\boldsymbol{x}_{t+1})\|^2 + \frac{4\gamma_t^3 L^2}{(1-\beta)^2}\mathbb{E}\|\boldsymbol{m}_t\|^2 + \frac{4\gamma_t^3 L^2}{(1-\beta)^2}\mathbb{E}\|\overline{\boldsymbol{\delta}}_{t-1}\|^2 + \frac{L\gamma_t^2}{2}\mathbb{E}\|\boldsymbol{g}_{t+1}\|^2 \quad (r_{threshold} < 1)$$

$$\leq -\frac{\gamma_t}{2}\mathbb{E}\|\nabla f(\boldsymbol{x}_{t+1})\|^2 + \frac{4\gamma_t^3 L^2}{(1-\beta)^2}\mathbb{E}\|\boldsymbol{m}_t\|^2 + \frac{4\gamma_t^3 \epsilon^2}{(1-\beta)^2} + \frac{L\gamma_t^2}{2}\mathbb{E}\|\boldsymbol{g}_{t+1}\|^2$$

$$\leq -\frac{\gamma_t}{2}\mathbb{E}\|\nabla f(\boldsymbol{x}_{t+1})\|^2 + \frac{4\gamma_t^3 L^2}{(1-\beta)^2}\mathbb{E}\|\boldsymbol{m}_t\|^2 + \frac{4\gamma_t^3 \epsilon^2}{(1-\beta)^2} + \frac{L\gamma_t^2}{2}\mathbb{E}\|\nabla f(\boldsymbol{x}_{t+1})\|^2 + \frac{L\gamma_t^2 \sigma^2}{2n}.$$

Summing up the equation above from $t=0$ to $t=T$ we get

$$\mathbb{E}f(\boldsymbol{y}_{T+1}) - \mathbb{E}f(\boldsymbol{y}_0) \leq \sum_{t=0}^{T} -\frac{(1-\gamma_t L)\gamma_t}{2}\mathbb{E}\|\nabla f(\boldsymbol{x}_t)\|^2 + \sum_{t=0}^{T}\frac{4\gamma_t^3 L^2}{(1-\beta)^2}\mathbb{E}\|\boldsymbol{m}_t\|^2 + \frac{4L^2\epsilon^2\sum_{t=0}^{T}\gamma_t^3}{(1-\beta)^2} + \frac{L\sigma^2\sum_{t=0}^{T}\gamma_t^2}{2n}.$$

Therefore if we $\gamma_t < \frac{1}{2L}$ and it's within certain range $\gamma_t \in [\gamma_{\min}, \gamma_{\max}]$, the equation above leads to

$$(1 - \gamma_{\max}L)\sum_{t=0}^{T}\mathbb{E}\|\nabla f(\boldsymbol{x}_t)\|^2$$

$$\leq \frac{2\mathbb{E}f(\boldsymbol{y}_0) - 2\mathbb{E}f(\boldsymbol{y}_{T+1})}{\gamma_{\min}} + \frac{8\gamma_{\max}^3 L^2}{(1-\beta)^2\gamma_{\min}}\sum_{t=0}^{T}\mathbb{E}\|\boldsymbol{m}_t\|^2 + \frac{8\gamma_{\max}^3 L^2\epsilon^2 T}{(1-\beta)^2\gamma_{\min}} + \frac{L\gamma_{\max}^2\sigma^2 T}{n\gamma_{\min}}. \quad (8)$$

Notice that we have

$$\boldsymbol{m}_t = (1-\beta)\sum_{s=0}^{t}\beta^{t-s}\overline{\boldsymbol{g}}_s + \sum_{s=0}^{t}\beta^{t-s}(\overline{\boldsymbol{\delta}}_{s-1} - \overline{\boldsymbol{\delta}}_s)$$

which by using Lemma 1, we have

$$\sum_{t=0}^{T}\|\boldsymbol{m}_t\|^2 \le \sum_{t=0}^{T}\|\boldsymbol{g}_t\|^2 + \frac{2}{(1-\beta)^2}\sum_{t=0}^{T}\|\overline{\boldsymbol{\delta}}_t\|^2 \le \sum_{t=0}^{T}\|\nabla f(\boldsymbol{x}_t)\|^2 + \frac{\sigma^2 T}{n} + \frac{2\epsilon^2 T}{(1-\beta)^2}. \quad (9)$$

Combing (8) and (9) together we get

$$\left(1 - \gamma_{\max}L - \frac{8\gamma_{\max}^3 L^2}{(1-\beta)^2\gamma_{\min}}\right)\sum_{t=0}^{T}\mathbb{E}\|\nabla f(\boldsymbol{x}_t)\|^2$$

$$\le\frac{2\mathbb{E}f(\boldsymbol{y}_0) - 2\mathbb{E}f(\boldsymbol{y}_{T+1})}{\gamma_{\min}} + \frac{24\gamma_{\max}^3 L^2\epsilon^2 T}{(1-\beta)^2\gamma_{\min}} + \frac{L\gamma_{\max}^2\sigma^2 T}{n\gamma_{\min}} + \frac{8\gamma_{\max}^3 L^2\sigma^2 T}{n(1-\beta)^2\gamma_{\min}}$$

$$\le\frac{2\mathbb{E}f(\boldsymbol{x}_1) - 2\mathbb{E}f(\boldsymbol{y}^*)}{\gamma_{\min}} + \frac{24\gamma_{\max}^3 L^2\epsilon^2 T}{(1-\beta)^2\gamma_{\min}} + \frac{L\gamma_{\max}^2\sigma^2 T}{n\gamma_{\min}} + \frac{8\gamma_{\max}^3 L^2\sigma^2 T}{n(1-\beta)^2\gamma_{\min}}.$$

$\square$

**Proof to Theorem 1**   Since using a per-coordinate learning rate for loss function $f(\cdot)$ is equivalent to use a constant learning for all coordinates but for loss function $h(\cdot)$, the only two thing that change are

- **Different L-Lipschitzian coefficient**: the L-Lipschitzian coefficient for $h(\cdot)$ is

$$\|\nabla h(\boldsymbol{x}) - \nabla h(\boldsymbol{y})\|^2 = \left\|V^{\frac{1}{2}}\nabla f(V^{\frac{1}{2}}\boldsymbol{x}) - V^{\frac{1}{2}}\nabla f(V^{\frac{1}{2}}\boldsymbol{y})\right\|^2$$

$$= \left\|\nabla f(V^{\frac{1}{2}}\boldsymbol{x}) - \nabla f(V^{\frac{1}{2}}\boldsymbol{y})\right\|_V^2$$

$$\le L^2\left\|V^{\frac{1}{2}}\boldsymbol{x} - V^{\frac{1}{2}}\boldsymbol{y}\right\|_V^2$$

$$= L^2\|\boldsymbol{x} - \boldsymbol{y}\|_{V^2}^2$$

$$\le L^2 V_{\max}^2\|\boldsymbol{x} - \boldsymbol{y}\|^2.$$

  Therefore the effective L-Lipschitzian coefficient of $h(\boldsymbol{x})$ is $LV_{\max}$

- **Different definition of $\overline{\boldsymbol{\delta}}_t$**: from (4) we shall see that actually the compression error in the view of $h(\cdot)$ is $V^{\frac{1}{2}}\overline{\boldsymbol{\delta}}_t$, so in this case we have

$$\mathbb{E}\|V^{\frac{1}{2}}\overline{\boldsymbol{\delta}}_t\|^2 \le V_{\max}\epsilon^2$$

*Proof.*  From Lemma 2, we have

$$\left(1 - \gamma_{\max}LV_{\max} - \frac{8\gamma_{\max}^3 L^2}{(1-\beta)^2\gamma_{\min}}\right)\sum_{t=0}^{T}\mathbb{E}\|\nabla f(\boldsymbol{x}_t)\|^2$$

$$\le\frac{2\mathbb{E}f(\boldsymbol{x}_1) - 2\mathbb{E}f(\boldsymbol{y}^*)}{\gamma_{\min}} + \frac{24\gamma_{\max}^3 L^2 V_{\max}^3\epsilon^2 T}{(1-\beta)^2\gamma_{\min}} + \frac{LV_{\max}\gamma_{\max}^2\sigma^2 T}{n\gamma_{\min}} + \frac{8\gamma_{\max}^3 L^2 V_{\max}^2\sigma^2 T}{n(1-\beta)^2\gamma_{\min}}.$$

$\square$

### A.9.3   PROOF TO COROLLARY 1

*Proof.*  By choosing $\gamma_{\max} = \gamma_{\min} = \frac{1-\beta}{16LV_{\max} + \sigma\sqrt{\frac{T}{n}} + T^{\frac{1}{3}}\epsilon^{\frac{2}{3}}}$, we can guarantee that

$$1 - \gamma L - \frac{8\gamma^2 L^2 V_{\max}^2}{(1-\beta)^2} \ge \frac{1}{2}.$$

So (3) leads to

$$
\sum_{t=0}^{T} \mathbb{E}\|\nabla f(\boldsymbol{x}_t)\|_V^2 \leq \frac{2\left(\mathbb{E}f(\boldsymbol{y}_0) - f(\boldsymbol{y}^*)\right)}{(1-\beta)}\left(16LV_{\max} + \sigma\sqrt{\frac{T}{n}} + T^{\frac{1}{3}}\epsilon^{\frac{2}{3}}\right)
$$

$$
+ \left((1-\beta)L\sqrt{T} + 2L^2 V_{\max}^2\right)\frac{\sigma}{\sqrt{n}} + 32L^2\epsilon^{\frac{2}{3}}T^{\frac{1}{3}}V_{\max}^3
$$

$$
\frac{1}{T}\sum_{t=0}^{T} \mathbb{E}\|\nabla f(\boldsymbol{x}_t)\|_V^2 \leq \frac{2\left(\mathbb{E}f(\boldsymbol{y}_0) - f(\boldsymbol{y}^*)\right)}{(1-\beta)}\left(\frac{16LV_{\max}}{T} + \frac{\sigma}{\sqrt{nT}} + T^{-\frac{2}{3}}\epsilon^{\frac{2}{3}}\right)
$$

$$
+ \left((1-\beta)L + \frac{2L^2 V_{\max}^2}{\sqrt{T}}\right)\frac{\sigma}{\sqrt{nT}} + 32L^2\epsilon^{\frac{2}{3}}T^{-\frac{2}{3}}V_{\max}^3.
$$

Treating $f(\boldsymbol{y}_1) - f^*$, $\beta$ and $L$ as constants, from the inequality above we get

$$
\frac{1}{T}\sum_{t=0}^{T} \mathbb{E}\|\nabla f(\boldsymbol{x}_t)\|^2 \lesssim \frac{\sigma}{\sqrt{nT}} + \frac{\epsilon^{\frac{2}{3}}}{T^{\frac{2}{3}}} + \frac{1}{T}.
$$

It completes the proof. $\qquad\square$

