# OpenReview forum: "1-bit LAMB: Communication Efficient Large-Scale Large-Batch Training with LAMB's Convergence Speed"
_ICLR.cc/2022/Conference — ICLR 2022 Submitted_

### Official Review · Reviewer_amJh · 2021-11-01

**Correctness:** 4
**Technical Novelty And Significance:** 2
**Empirical Novelty And Significance:** 2
**Recommendation:** 5
**Confidence:** 4

**Main Review:**

Strengths

1. An interesting insight that the norm of LAMB's variance is more unstable than Adam

2. Comprehensive experimental results to validate the claims, including the motivation and proposed method.

Weaknesses

1. Given the prior work 1-bit Adam, the novelty becomes somewhat incremental. The novel part is that 1-bit LAMB will try to adjust the scaling coefficient of LAMB. The efficient communication of momentum follows existing communication-efficient momentum SGD method.

The reconstruction of the gradient means that other compressors (e.g., Top-K) may not be incorporated into the proposed method.

2. Does 1-bit LAMB preserve the same performance as LAMB with different number of workers and batch size? It would be great to have a table summarizing the performance with different #worker and batch size settings. Figure 3 only gives the training throughput.

3. The theoretical analysis does not seem to validate the coefficient scaling in 1-bit LAMB. It may need further heuristic and theoretical elaboration.

4. The 3rd assumption in the theoretical analysis does not seem very common to me in communication-efficient distributed training. The author needs to validate it. A more commonly-used assumption about the compressor is $||C(x)-x||^2 \leq (1-\delta)||x||^2$, from which we may know more about the effect of the compression ratio. Therefore, the convergence analysis of 1-bit LAMB seems less solid to me.

**Summary Of The Paper:**

The paper proposes a communication-efficient distributed LAMB optimizer with 1-bit compression. It follows previous work to first warm-up the variance, but proposes to inference the scaling factor based on reconstructed variance. Experiments show training speedup due to communication compression. The proposed 1-bit LAMB achieves similar model performance to full-precision LAMB.

**Summary Of The Review:**

The paper propose a communication-efficient large-batch LAMB optimizer to accelerate distributed training. The experimental results looks promising, but there could be some improvements both empirically and theoretically.

---

> ### Author Response · Authors · 2021-11-23
> **Rebuttal reply**
>
> Thank you for your comments and below are our replies to some of them. In addition, in the rebuttal paper revision we added some new experiments/analysis related to the comments from reviewers: 1) Appendix A.7 where we provide additional BERT/SQuAD convergence analysis with different number of workers and different batch sizes. 2) Appendix A.8 where we provide additional convergence analysis on CIFAR100.
>
> <Comment 1> "Given the prior work 1-bit Adam, the novelty becomes somewhat incremental. The novel part is that 1-bit LAMB will try to adjust the scaling coefficient of LAMB. The efficient communication of momentum follows existing communication-efficient momentum SGD method. The reconstruction of the gradient means that other compressors (e.g., Top-K) may not be incorporated into the proposed method."
>
> <Reply 1> For the reconstruction of the gradient, we agree that using Top-K might cause some dimension of the gradient to be zero and influence the convergence of our algorithm. We agree that our 1-bit LAMB work adapts the 2-stage strategy proposed in 1-bit Adam, but we politely argue that our design of the adaptive scaling based on gradient recovery and stabilizing mechanisms is a novel and solid contribution based on many different trials. We tried many strategies: (1) Only freeze variance; (2) Freeze both variance and LAMB scaling coefficients; (3) Freeze both variance and LAMB scaling coefficients, and multiply a fixed factor to the frozen LAMB coefficients; (4) Freeze both variance and LAMB scaling coefficients, and multiply a linearly increasing factor to the frozen LAMB coefficients. At the end we finally settled to an adaptive and efficient design presented in our paper. To our best knowledge, this is the first work that could apply communication compression to large batch-size training and achieve both superior training efficiency and on-par/better convergence accuracy.
>
> <Comment 2> "Does 1-bit LAMB preserve the same performance as LAMB with different number of workers and batch size? It would be great to have a table summarizing the performance with different \#worker and batch size settings. Figure 3 only gives the training throughput."
>
> <Reply 2> In the rebuttal paper revision we added Appendix A.7 where we compare the convergence of LAMB and 1-bit LAMB under different number of workers and different batch sizes. Results show that 1-bit LAMB provides on-par/better SQuAD F1 scores than LAMB at all cases.
>
> <Comment 3> "The theoretical analysis does not seem to validate the coefficient scaling in 1-bit LAMB. It may need further heuristic and theoretical elaboration."
>
> <Reply 3> We agree that our theoretical analysis does not prove the superiority of using the scaling strategy in 1-bit LAMB, because it is a very challenging work. Therefore we finally settled into the current analysis. We've  checked the original LAMB work, and found that their results also indicates that LAMB's scaling coefficient will lead to a slightly worse convergence rate. We believe this is a very interesting question and we will keep investigate this in our future work.
>
> <Comment 4> "The 3rd assumption in the theoretical analysis does not seem very common to me in communication-efficient distributed training. The author needs to validate it. A more commonly-used assumption about the compressor is $\|\|C(x) - x\|\|^2 \leq (1 - \delta)\|\|x\|\|^2$
> , from which we may know more about the effect of the compression ratio. Therefore, the convergence analysis of 1-bit LAMB seems less solid to me."
>
> <Reply 4> The most important reason we use this assumption is that for 1-bit compression, the signal-to-noise assumption does not hold if there is only one non-zero element in the tensor. Additionally, in related work [1] Section 4 authors proved that our assumption is equivalent to the signal-to-noise assumption when the gradient magnitude is upper bounded. We added this clarification in the rebuttal paper revision.
>
> [1] Hanlin Tang, Xiangru Lian, Chen Yu, Tong Zhang, Ji Liu. “DoubleSqueeze: Parallel Stochastic Gradient Descent with Double-Pass Error-Compensated Compression.” ICML (2019).

---

### Official Review · Reviewer_otsu · 2021-11-02

**Correctness:** 3
**Technical Novelty And Significance:** 3
**Empirical Novelty And Significance:** 3
**Recommendation:** 5
**Confidence:** 3

**Main Review:**

The paper is well-written. The idea of 1-bit lamb is well-supported by the convergence analysis and good empirical results. In overall, I think this is a good paper.
Here are my concerns:
1. All the experiments are limited to BERT, which makes it unknown whether the proposed algorithms could be applied to other training tasks including cv and nlp. To justify the effectiveness of the proposed algorithm, I recommend to add at least 1 CV task, such as resnet on cifar100 or imagenet.
2. My major concern is in the theoretical analysis. Some part of the theoretical analysis is confusing to me. I cannot find the definition of $V_{max}$, so I simply guess it is the maximum value of $V$. However, since the diagonal elements of $V$ are actually $1/v$, $V_{max}$ is actually assuming a lower bound of the gradient $\|\nabla f(x)\|$. Such a lower bound seems unreasonable to me since the convergence of the algorithm actually means that $\|\nabla f(x)\|$ should approach 0, which contradicts the lower bound of $\|\nabla f(x)\|$. Could the authors resolve such an issue?

**Summary Of The Paper:**

In this paper, the authors propose 1-bit LAMB, which combines lamb optimizer with communication compression, with convergence analysis and experiment results.

**Summary Of The Review:**

The paper is well-written. The idea of 1-bit lamb is well-supported by the convergence analysis and good empirical results. In overall, I think this is a good paper. Some other experiments is recommended. The main issue is in the assumption of the theoretical analysis.

---

> ### Author Response · Authors · 2021-11-23
> **Rebuttal reply**
>
> Thank you for your comments and below are our replies to some of them. In addition, in the rebuttal paper revision we added some new experiments/analysis related to the comments from reviewers: 1) Appendix A.7 where we provide additional BERT/SQuAD convergence analysis with different number of workers and different batch sizes. 2) Appendix A.8 where we provide additional convergence analysis on CIFAR100.
>
> <Comment 1> "All the experiments are limited to BERT, which makes it unknown whether the proposed algorithms could be applied to other training tasks including cv and nlp. To justify the effectiveness of the proposed algorithm, I recommend to add at least 1 CV task, such as resnet on cifar100 or imagenet."
>
> <Reply 1> We agree and in the rebuttal paper revision we added a ResNet-50 on CIFAR100 experiment in Appendix A.8. Results show that 1-bit LAMB provides on-par/better testing accuracy than LAMB.
>
> <Comment 2> "My major concern is in the theoretical analysis. Some part of the theoretical analysis is confusing to me. I cannot find the definition of $V_{\max}$, so I simply guess it is the maximum value of $V$. However, since the diagonal elements of $V$ are actually $1/v$, $V_{\max}$ is actually assuming a lower bound of the gradient $\|\|\nabla f(x)\|\|$. Such a lower bound seems unreasonable to me since the convergence of the algorithm actually means that $\|\|\nabla f(x)\|\|$ should approach 0, which contradicts the lower bound of $\|\|\nabla f(x)\|\|$. Could the authors resolve such an issue?"
>
> <Reply 2> We politely argue that here $1/v$ is a moving sum over all history $\left(\nabla f(x)\right)^2$, not $\|\|\nabla f(x)\|\|$. Therefore it will not approach to $0$ for the following two reasons: i) We stop updating $v$ after the warmup stage, which means $\nabla f(x)$ at this time will not approach $0$ since it has not converged yet; ii) Since $\beta_2=0.999$, this means $v$ is heavily dependent to the history gradients, therefore even though $\nabla f(x)$ approaches to $0$, $v$ itself will still taking a time to decay to $0$. Therefore it is reasonable for us to assume an upper bound for $V_{\max}$. We added this clarification in the rebuttal paper revision.

---

### Official Review · Reviewer_5SMi · 2021-11-08

**Correctness:** 4
**Technical Novelty And Significance:** 2
**Empirical Novelty And Significance:** 3
**Recommendation:** 6
**Confidence:** 3

**Main Review:**

Strengths:
* In general, the results demonstrated in the work show measurable improvements in efficiency over the baselines: the gains of 2-3x might be quite significant for practical scenarios of large-scale learning. Authors also compare their method with several reasonably chosen baselines to demonstrate the importance of their contributions: while there definitely could be more pretraining tasks and models to be considered, given the space limits I believe the current scope of experiments is sufficient to justify the claims made in the paper.
* Moreover, the theoretical analysis of the algorithm allows to validate its scalability properties and to establish convergence rates mostly equivalent to those seen in previous works on distributed optimization.

Weaknesses:
* In the text, I could not find any information about the actual compression algorithm used for experiments. I think this is a quite important part of the algorithm which should be described in detail, since both the empirical results and the analysis (Assumption 1.3) rely on it.
* Although the simplicity of the method is definitely a plus, some parts of the contribution of this work appear to lack any significant novelty. For instance, it was not clear to me how the primitive described in Section 5 differs from regular All-Reduce and whether this contribution is solely a software engineering one (that is, porting the existing protocol from MPI to NCCL). The method also seems to generally follow from 1-bit Adam, with the exception of a gradient recovery mechanism (although authors indeed demonstrate that direct application of the techniques from prior work yields worse results).
* The presentation of the method can be slightly improved: for instance, is it necessary to mention a server as a separate entity if in practice all communication happens by means of All-Reduce?

Questions:
* Do I understand correctly that Algorithm 1 mostly reduces to 1-bit Adam if we remove all lines required to update $c_t^{(l)}$ (in particular, lines 15–20) and change LAMB to ADAM for the preconditioning phase?

**Summary Of The Paper:**

This work studies the problem of distributed training with large batches in a communication-bottlenecked setup, where regular versions of algorithms such as LAMB become a constraint. Authors propose a new algorithm, which compresses the gradient momentum before aggregation and then reconstructs the gradients to recover the scaling coefficients for LAMB. This idea allows authors to achieve faster convergence compared to naive compression for LAMB while maintaining higher communication efficiency than a non-compressed version.

**Summary Of The Review:**

Although this paper proposes a straightforward yet definitely useful method for speeding up distributed training, in my opinion the clarity of presentation both for method and results can be improved. I give a score of a weak accept for now, with possible improvements if authors address the issues outlined in the review.

---

> ### Author Response · Authors · 2021-11-23
> **Rebuttal reply**
>
> Thank you for your comments and below are our replies to some of them. In addition, in the rebuttal paper revision we added some new experiments/analysis related to the comments from reviewers: 1) Appendix A.7 where we provide additional BERT/SQuAD convergence analysis with different number of workers and different batch sizes. 2) Appendix A.8 where we provide additional convergence analysis on CIFAR100.
>
> <Comment 1> "In the text, I could not find any information about the actual compression algorithm used for experiments. I think this is a quite important part of the algorithm which should be described in detail, since both the empirical results and the analysis (Assumption 1.3) rely on it."
>
> <Reply 1> Yes it is true that in the paper we use $C_{\omega}(\cdot)$ to denote the compressor but didn't define it for the case of 1-bit compression. In the rebuttal paper revision we added the definition when using 1-bit compression at Algorithm 1 line 1: $C_{\omega}[x] = \frac{\|\|x\|\|}{\|\|\text{Sign}(x)\|\|}\text{Sign}(x)$ where $\text{Sign}(x)$ is the tensor that only keeps the sign of each element in the original tensor $x$.
>
> <Comment 2> "Although the simplicity of the method is definitely a plus, some parts of the contribution of this work appear to lack any significant novelty. For instance, it was not clear to me how the primitive described in Section 5 differs from regular All-Reduce and whether this contribution is solely a software engineering one (that is, porting the existing protocol from MPI to NCCL). The method also seems to generally follow from 1-bit Adam, with the exception of a gradient recovery mechanism (although authors indeed demonstrate that direct application of the techniques from prior work yields worse results)."
>
> <Reply 2> For the primitive described in Section 5, we call it a "compressed allreduce" but it is a non-standard custom communication operation. We use computation kernels for compression of the data passed to this primitive and then use other collective communication operations like gather and alltoall to implement an all-reduce like behavior. We note that standard all-reduce is usually an element-wise summation or averaging of the input data whereas the proposed primitive is compression of the input data as well as the averaging step. The NCCL-based implementation proposed in this paper is a non-trivial engineering effort following the protocol from 1-bit Adam paper's MPI-based implementation. This NCCL-based implementation provides much better usability and better performance than MPI-based implementation as described in Appendix A.6.
>
> We agree that our 1-bit LAMB work adapts the 2-stage strategy proposed in 1-bit Adam, but we politely argue that our design of the adaptive scaling based on gradient recovery and stabilizing mechanisms is a novel and solid contribution based on many different trials. We tried many strategies: (1) Only freeze variance; (2) Freeze both variance and LAMB scaling coefficients; (3) Freeze both variance and LAMB scaling coefficients, and multiply a fixed factor to the frozen LAMB coefficients; (4) Freeze both variance and LAMB scaling coefficients, and multiply a linearly increasing factor to the frozen LAMB coefficients. At the end we finally settled to an adaptive and efficient design presented in our paper. To our best knowledge, this is the first work that could apply communication compression to large batch-size training and achieve both superior training efficiency and on-par/better convergence accuracy.
>
> <Comment 3> "The presentation of the method can be slightly improved: for instance, is it necessary to mention a server as a separate entity if in practice all communication happens by means of All-Reduce?"
>
> <Reply 3> We believe that the concept of server is helpful to explain the two-stage compressed all-reduce algorithm in a more general way: first workers compress and send to server, then server averages, compresses again, and sends back to workers. It is true that in our actual implementation each worker acts as the server as well (In the rebuttal paper revision we added additional clarification in Section 5.), but in other possible implementations server could be a separate entity.
>
> <Comment 4> "Do I understand correctly that Algorithm 1 mostly reduces to 1-bit Adam if we remove all lines required to update $c_t^{(l)}$ (in particular, lines 15–20) and change LAMB to ADAM for the preconditioning phase?"
>
> <Reply 4> Yes this is correct, since 1-bit LAMB and 1-bit Adam have a similar relationship as LAMB and Adam.

---

### Decision · Program_Chairs · 2022-01-20

**Decision:**

Reject

**Comment:**

The authors propose a communication-efficient distributed LAMB optimizer using a 1-bit compression. This work is similar in spirit to other prior work, eg 1-bit Adam. Although the algorithm works reasonably well it is a bit unclear how much compression is achieved. Overall, the algorithmic novelty is limited, given the prior work, and the benefits of the algorithm don't shine through as the experiments are quite limited in their data sets and models. The theoretical results are also of unclear usefulness due to the assumptions made.